# Maximizing CRISPRi efficacy and accessibility with dual-sgRNA libraries and optimal effectors

Joseph M Replogle[1,2,3,4†], Jessica L Bonnar[2,3,4†‡], Angela N Pogson[2,3,4§], Christina R Liem[2#], Nolan K Maier[5], Yufang Ding[5], Baylee J Russell[5], Xingren Wang[5], Kun Leng[1,6], Alina Guna[2,4], Thomas M Norman[2¶], Ryan A Pak[2**], Daniel M Ramos[7,8], Michael E Ward[9], Luke A Gilbert[2,10,11], Martin Kampmann[6,12], Jonathan S Weissman[2,3,4,13,14]*, Marco Jost[2,5]*

[1]Medical Scientist Training Program, University of California, San Francisco, San Francisco, United States; [2]Department of Cellular and Molecular Pharmacology, University of California, San Francisco, San Francisco, United States; [3]Howard Hughes Medical Institute, Massachusetts Institute of Technology, Cambridge, United States; [4]Whitehead Institute for Biomedical Research, Cambridge, United States; [5]Department of Microbiology, Harvard Medical School, Boston, United States; [6]Institute for Neurodegenerative Disease, University of California, San Francisco, San Francisco, United States; [7]Center for Alzheimer's Disease and Related Dementias, National Institutes of Health, Bethesda, United States; [8]National Institute on Aging, National Institutes of Health, Bethesda, United States; [9]National Institute of Neurological Disorders and Stroke, National Institutes of Health, Bethesda, United States; [10]Department of Urology and Helen Diller Family Comprehensive Cancer Center, University of California, San Francisco, San Francisco, United States; [11]Arc Institute, Palo Alto, United States; [12]Department of Biochemistry and Biophysics, University of California, San Francisco, San Francisco, United States; [13]Department of Biology, Massachusetts Institute of Technology, Cambridge, United States; [14]David H. Koch Institute for Integrative Cancer Research, Massachusetts Institute of Technology, Cambridge, United States

*For correspondence: weissman@wi.mit.edu (JSW); marco_jost@hms.harvard.edu (MJ)

[†]These authors contributed equally to this work

Present address: [‡]Department of Biology, Massachusetts Institute of Technology, Cambridge, United States; [§]Department of Developmental Biology, Stanford University School of Medicine, Stanford, United States; [#]Division of Biological Sciences, Section of Cell and Developmental Biology, University of California San Diego, San Diego, United States; [¶]Computational & Systems Biology Program, Sloan Kettering Institute, Memorial Sloan Kettering Cancer Center, New York, United States; [**]Department of Neuroscience, Scripps Research, La Jolla, United States

**Abstract** CRISPR interference (CRISPRi) enables programmable, reversible, and titratable repression of gene expression (knockdown) in mammalian cells. Initial CRISPRi-mediated genetic screens have showcased the potential to address basic questions in cell biology, genetics, and biotechnology, but wider deployment of CRISPRi screening has been constrained by the large size of single guide RNA (sgRNA) libraries and challenges in generating cell models with consistent CRISPRi-mediated knockdown. Here, we present next-generation CRISPRi sgRNA libraries and effector expression constructs that enable strong and consistent knockdown across mammalian cell models. First, we combine empirical sgRNA selection with a dual-sgRNA library design to generate an ultra-compact (1–3 elements per gene), highly active CRISPRi sgRNA library. Next, we compare CRISPRi effectors to show that the recently published Zim3-dCas9 provides an excellent balance between strong on-target knockdown and minimal non-specific effects on cell growth or the transcriptome. Finally, we engineer a suite of cell lines with stable expression of Zim3-dCas9 and robust on-target knockdown. Our results and publicly available reagents establish best practices for CRISPRi genetic screening.

---

## Editor's evaluation

Replogle et al. present their design of a compact and functionally validated dual sgRNA library and dCas9-effector protein that will enable new forms of CRISPRi-based screening in mammalian cells. Quantitative comparisons to previously published standards demonstrate strengths and weaknesses that along with the protocols and design strategies outlined, should enable end-users to rapidly adopt their approach.

## Introduction

CRISPR interference (CRISPRi) enables programmable repression of gene expression with broad applications in genome engineering, genetic screening, and cell biology (*Doench, 2018*). In mammalian cells, CRISPRi requires two components: (i) an effector protein of catalytically dead Cas9 (dCas9) fused to one or more transcription repressor domains, which recruits endogenous epigenetic modulators to the genome, and (ii) a single guide RNA (sgRNA), which directs the effector protein to target DNA (*Gilbert et al., 2013*). When the sgRNA is targeted to a gene promoter, CRISPRi leads to repressive epigenome editing and knockdown of the gene (*Gilbert et al., 2014*; *Horlbeck et al., 2016a*, *Horlbeck et al., 2016b*).

Several features distinguish CRISPRi from Cas9 nuclease-mediated DNA cutting, the major alternative CRISPR/Cas-based approach for loss-of-function genetic studies: (i) Unlike Cas9, CRISPRi does not rely on introduction of double-stranded DNA breaks and therefore does not cause genomic rearrangements (*Kosicki et al., 2018*) and DNA damage-associated toxicity (*Meyers et al., 2017*), which may be especially limiting in primary and stem cells (*Bowden et al., 2020*; *Haapaniemi et al., 2018*; *Ihry et al., 2018*). (ii) CRISPRi tends to confer more homogeneous loss of gene function compared to Cas9, which often generates subpopulations of cells bearing active in-frame indels (*Smits et al., 2019*). (iii) CRISPRi is reversible and thus affords temporal control over gene expression levels (*Gilbert et al., 2014*; *Mandegar et al., 2016*). (iv) CRISPRi enables titration of gene expression, which for example allows for partial depletion of genes essential for cell growth and interrogation of the resulting phenotypes (*Bosch et al., 2021*; *Hawkins et al., 2020*; *Jost et al., 2020*). (v) In turn, one can directly measure the extent of on-target knockdown as well as the corresponding responses in individual cells, for example, using single-cell RNA-seq (Perturb-seq), allowing for evaluation of the extent and potential biological significance of cell-to-cell heterogeneity. (vi) CRISPRi enables loss-of-function studies for non-coding RNAs, which are difficult to inactivate or repress through CRISPR cutting and the introduction of indels as they are insensitive to frame-shifting mutations (*Liu et al., 2017*).

Like other CRISPR approaches, CRISPRi has been paired with large-scale sgRNA libraries to conduct systematic genetic screens. Such screens have been deployed to identify essential protein-coding and non-coding genes (*Gilbert et al., 2014*; *Haswell et al., 2021*; *Horlbeck et al., 2016a*; *Liu et al., 2017*; *Raffeiner et al., 2020*), to map the targets of regulatory elements (*Fulco et al., 2019*; *Fulco et al., 2016*; *Gasperini et al., 2019*; *Kearns et al., 2015*; *Klann et al., 2017*; *Thakore et al., 2015*), to identify regulators of cellular signaling and metabolism (*Coukos et al., 2021*; *Liang et al., 2020*; *Luteijn et al., 2019*; *Semesta et al., 2020*), to uncover stress response pathways in stem cell-derived neurons (*Tian et al., 2021*; *Tian et al., 2019*), to uncover regulators of disease-associated states in microglia and astrocytes (*Dräger et al., 2022*; *Leng et al., 2022*), to decode regulators of cytokine production in primary human T-cells (*Schmidt et al., 2022*), to define mechanisms of action of bioactive small molecules (*Jost et al., 2017*; *Morgens et al., 2019*; *le Sage et al., 2017*), to identify synthetic-lethal genetic interactions in cancer cells (*Du et al., 2017*; *Horlbeck et al., 2018*), and to identify genetic determinants of complex transcriptional responses using RNA-seq readouts (Perturb-seq) (*Adamson et al., 2016*; *Replogle et al., 2022*; *Replogle et al., 2020*; *Tian et al., 2021*; *Tian et al., 2019*), among others.

Despite these successes, two technical factors have limited wider adoption of CRISPRi. First, CRISPRi screening is constrained by the large size of sgRNA libraries. Previous machine learning efforts yielded guide design rules which substantially increased the activity of sgRNA libraries (*Horlbeck et al., 2016a*; *Sanson et al., 2018*). Nonetheless, commonly used libraries (e.g., Dolcetto, CRISPRi v2) target each gene with three or more sgRNAs to decrease false-negative results in screens. The development of a more compact, highly active sgRNA library would enable CRISPRi screens in new cell types and for more complex phenotypes, especially when cost, time, and/or cell numbers are

limiting. Second, there is no clear consensus guiding the use of the different reported CRISPRi effector proteins, complicating the generation of CRISPRi cell models (*Alerasool et al., 2020*; *Carleton et al., 2017*; *Gilbert et al., 2014*; *Yeo et al., 2018*).

Here, we present a suite of tools to enable high-quality CRISPRi genetic screening in a broad range of cell models as well as accompanying protocols, publicly available at https://www.jostlab.org/resources/ and https://weissman.wi.mit.edu/resources/. Based on empirical data aggregated from 126 screens, we design and validate an ultra-compact, highly active CRISPRi library in which each gene is targeted by a single library element encoding a dual-sgRNA cassette of the two most active sgRNAs for that gene. Next, we compare published CRISPRi effector proteins based on their on-target efficacy and non-specific effects on transcription and cell proliferation. We find that the recently published Zim3-dCas9 provides the best balance between strong on-target knockdown and minimal non-specific effects. Finally, we generate K562, RPE1, Jurkat, HT29, HuTu-80, and HepG2 cell lines engineered to stably express Zim3-dCas9 and demonstrate robust on-target knockdown across these cell lines. Our results and reagents establish best practices for CRISPRi genetic screening.

## Results

### Comparison of single- and dual-sgRNA CRISPRi libraries for genetic screening

Two critical factors for potential applications of CRISPRi screening are the on-target knockdown efficacy and the size of the sgRNA library. In recent work, we found that targeting individual genes with dual-sgRNA constructs substantially improved CRISPRi-mediated gene knockdown (*Replogle et al., 2020*). Building on this result, we asked whether a dual-sgRNA strategy could be used to generate an ultra-compact, genome-wide CRISPRi library.

To assess the potential utility of dual-sgRNA libraries in systematic genetic screens, we began by cloning two pilot libraries for comparison: (i) one targeting each human gene with two distinct sgRNAs expressed from a tandem sgRNA cassette (dual-sgRNA) and (ii) one targeting each human gene by only the single best sgRNA (see Materials and methods; *Supplementary file 1*). We also optimized a protocol to amplify and sequence dual-sgRNA cassettes from lentivirally integrated genomic DNA (see Materials and methods; both protocols available at https://www.jostlab.org/resources/ and https://weissman.wi.mit.edu/resources/). Next, we compared the performance of our single- and dual-sgRNA libraries in a genome-wide growth screen (*Figure 1A*). We transduced K562 cells stably expressing dCas9-KRAB(Kox1) with our libraries, used puromycin to select for cells with lentiviral integration, and harvested cells at day 8 ($T_0$) and day 20 ($T_{final}$) post-transduction. We amplified sgRNA cassettes from extracted genomic DNA, sequenced to quantify sgRNA abundance in the two populations, and calculated growth phenotypes for each library element by comparing changes in abundance between $T_0$ and $T_{final}$ (*Figure 1C*, *Supplementary file 2*). The growth phenotypes produced by the single- and dual-sgRNA libraries were well correlated with previously published CRISPRi growth screens using five sgRNAs per gene (single sgRNA r=0.82; dual sgRNA r=0.83; *Figure 1—figure supplement 1A–C*) and produced near-perfect recall of essential genes (*Figure 1—figure supplement 1D*) (AUC > 0.98 for both single- and dual-sgRNA libraries). Yet, for essential genes previously identified by the Cancer Dependency Map (DepMap) (*Behan et al., 2019*; *Tsherniak et al., 2017*), the dual-sgRNA library produced significantly stronger growth phenotypes (mean 29% decrease in the growth rate [γ]) than the single-sgRNA library (n=2005 genes; single-sgRNA mean γ = −0.20; dual-sgRNA mean γ = −0.26; Mann-Whitney p-value = $6 \cdot 10^{-15}$; *Figure 1C and D*), suggesting that the dual-sgRNA library confers stronger depletion of target genes.

A well-recognized challenge for the use of dual-sgRNA libraries is that the lentiviral reverse transcriptase can undergo template switching between the two copies of the lentiviral genome packaged into each capsid (*Adamson et al., 2018*; *Adamson et al., 2016*; *Feldman et al., 2018*; *Hill et al., 2018*; *Horlbeck et al., 2018*; *Xie et al., 2018*). These two copies generally bear two different sgRNA pairs in a pooled dual-sgRNA library, such that template switching can produce a recombined element with sgRNAs targeting different genes. Our sequencing strategy allowed us to directly identify such recombined elements (*Figure 1B*), which occurred with a frequency of 29% in K562 cells, 26% in RPE1 cells, and 24% in Jurkat cells, consistent with prior reports (*Horlbeck et al., 2018*; *Replogle et al., 2020*). We expected that recombination would be stochastic rather than biased to specific sgRNA

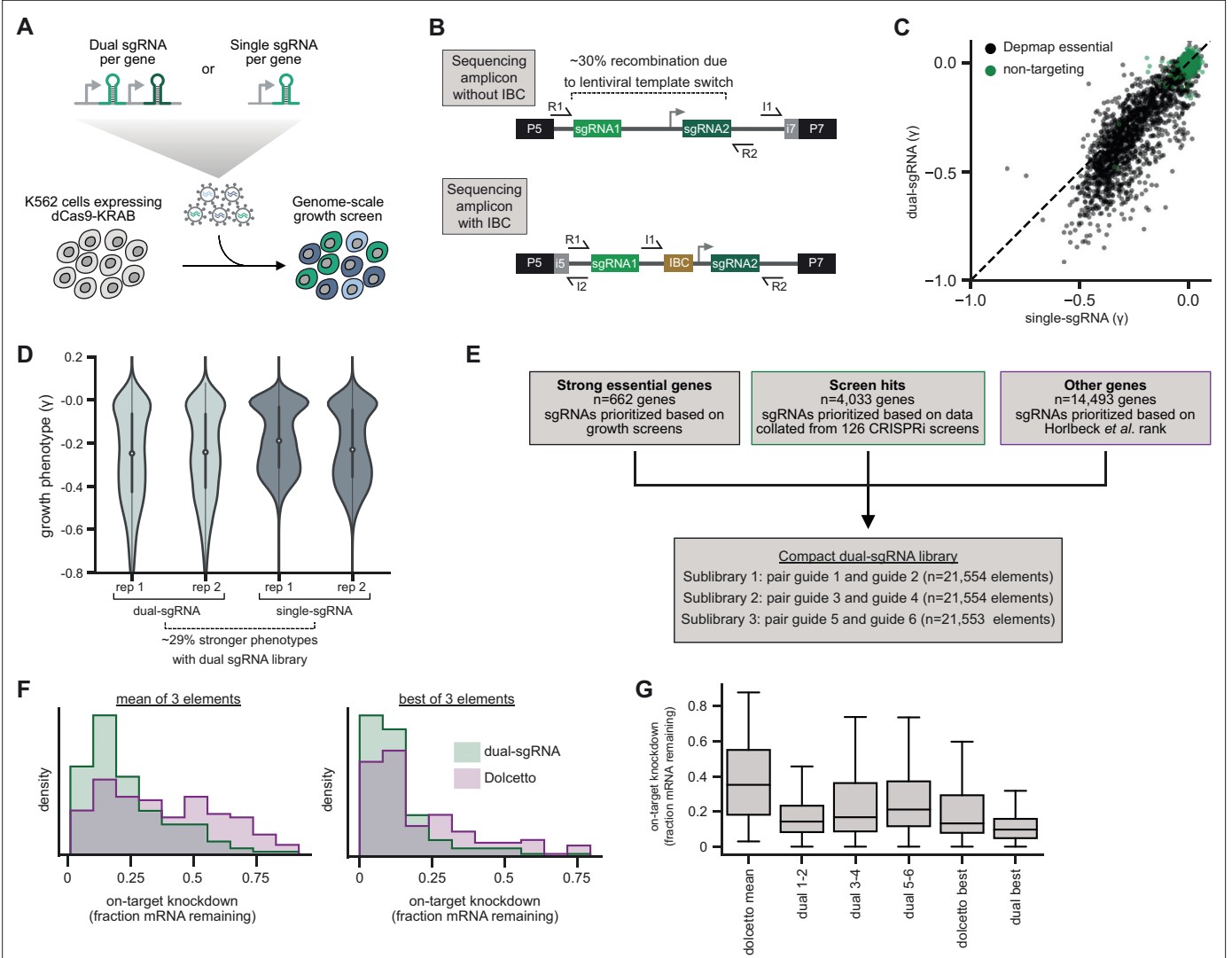

**Figure 1.** Design and validation of ultra-compact dual-single guide RNA (sgRNA) CRISPR interference (CRISPRi) libraries. (**A**) Schematic of growth screen used to compare single- and dual-sgRNA libraries. (**B**) Schematic of dual-sgRNA library sequencing strategies. (**C**) Comparison of growth phenotypes for DepMap essential genes between single- and dual-sgRNA libraries. Sequencing libraries were prepared using the strategy labeled 'Sequencing amplicon without IBC' in panel B. Growth phenotypes are reported as γ (log$_2$ fold-enrichment of T$_{final}$ over T$_0$, per doubling) and well correlated between libraries (r=0.91). Only values between –1 and 0.1 are shown. (**D**) Comparison of growth phenotypes for DepMap essential genes between single- and dual-sgRNA libraries. In the violin plot, the violin displays the kernel density estimate, the central white point represents the median, and the central black bar represents the interquartile range (IQR). (**E**) Design of final dual-sgRNA library. (**F**) Comparison of target gene knockdown by dual-sgRNA library versus Dolcetto library. Target gene knockdown was measured by single-cell RNA-sequencing (Perturb-seq). For each library, the 'mean of 3 elements' was calculated as the mean knockdown of all three elements targeting each gene. The 'best of 3 elements' represents the element with the best knockdown per each gene. (**G**) Comparison of target gene knockdown across elements in dual-sgRNA library versus Dolcetto. In the box plot, the box shows the IQR, the line dividing the box shows the median value, and the whiskers extend to show 1.5× the IQR. Outlier observations >1.5× IQR are not shown.

The online version of this article includes the following figure supplement(s) for figure 1:

**Figure supplement 1.** Additional comparisons of pilot single- and dual-single guide RNA (sgRNA) library screens.

sequences, because the length of the invariable sequence between sgRNAs (400 bp) far exceeds the length of sgRNA targeting regions (20 bp). To further distinguish between these possibilities, we compared the recombination rates of elements across replicate screens in K562 cells and found that recombination rates of non-targeting control elements were only weakly correlated (r=0.30, *Figure 1—figure supplement 1E*). We note that apparent recombination rates of targeting elements were strongly correlated across replicates (r=0.77, *Figure 1—figure supplement 1F*). This correlation

is likely a consequence of more rapid dropout of unrecombined elements, which generally confer stronger growth phenotypes than recombined elements, leading to inflation of apparent recombination rates. Indeed, apparent recombination rates were strongly anticorrelated with growth phenotypes of the unrecombined elements (r = −0.84, *Figure 1—figure supplement 1G*). Together, these results further indicate that recombination is stochastic and largely independent of sgRNA sequence. In our downstream analyses, we exclude all recombined elements such that they do not impact phenotypes, although in principle these recombined elements could be used to assess independent effects of the two sgRNAs targeting each gene. Python scripts to quantify and remove recombined reads are available at https://github.com/josephreplogle/CRISPRi-dual-sgRNA-screens (*Replogle, 2022*; copy archived at swh:1:rev:87883ec73404ed02acba4089cebdf6db2b8cc673).

## Design and validation of ultra-compact, dual-sgRNA CRISPRi libraries

Having validated the performance of dual-sgRNA libraries in a systematic genetic screen, we sought to optimize the activity and utility of dual-sgRNA CRISPRi libraries (*Figure 1E*). To optimize sgRNA selection for each gene, we aggregated empirical sgRNA activity data from 126 CRISPRi genetic screens (*Supplementary file 3*) and implemented a three-tiered selection system. First, for genes that are essential in K562 cells, we ranked sgRNAs by growth phenotype. Second, for genes that produced a significant phenotype in one of our previous CRISPRi screens, we ranked sgRNAs by relative *z*-scored phenotype averaged across screens in which the target gene was identified as a hit. Finally, for genes without any empirical effect in a prior screen, we ranked sgRNAs according to predicted activities from the hCRISPRi v2.1 algorithm (see Materials and methods) (*Horlbeck et al., 2016a*). To allow users to select the library size suitable to their application, we cloned sublibraries of the best single element (guide ranked 1+2; referred to as hCRISPRi_dual_1_2), the second best element (guides ranked 3+4; referred to as hCRISPRi_dual_3_4), or the third best element (guides ranked 5+6; referred to as hCRISPRi_dual_5_6) (*Supplementary file 4*).

Further examination of the phenotypes from our screens revealed that a small number of elements produced discordant effects between screens, which may arise from bottlenecking or amplification bias (*Figure 1—figure supplement 1A–C*). For libraries with multiple elements targeting each gene, such discordant effects can often be mitigated by comparing phenotypes across elements, but this option is not available with single-element libraries. In previously reported CRISPR cutting libraries, incorporation of barcodes into the sgRNA cassette enabled marking and tracing populations of cells derived from individual lentiviral integrations, which allowed for detection of bottlenecking events and amplification bias and thereby improved screen sensitivity and robustness (*Michlits et al., 2017*; *Zhu et al., 2019*). Building on these results, we incorporated a set of 215 8-nucleotide barcodes, which we term integration barcodes (IBCs), in the tandem sgRNA cassette of our final hCRISPRi_dual_1_2, hCRISPRi_dual_3_4, and hCRISPRi_dual_5_6 libraries (Materials and methods, *Supplementary file 5*). We then optimized a sequencing strategy for simultaneously sequencing the two sgRNAs, the IBC, and a sample index on Illumina sequencers (*Figure 1B*; protocol available at https://www.jostlab.org/resources/ and https://weissman.wi.mit.edu/resources/).

Finally, we sought to test our optimized dual-sgRNA library side-by-side with the recently reported Dolcetto CRISPRi library, which was designed with a differently prioritized sgRNA selection algorithm and uses single sgRNAs (*Sanson et al., 2018*). We used direct capture Perturb-seq (*Replogle et al., 2020*), pooled CRISPR screens with single-cell RNA-seq readout, to measure the on-target knockdown mediated by the top three elements in our dual-sgRNA library (guides 1+2, guides 3+4, or guides 5+6) or the three Dolcetto Set A sgRNAs for 128 randomly selected genes that are expressed in K562 cells (*Supplementary file 6*). Our dual-sgRNA library significantly outperformed the Dolcetto library, as quantified by the average knockdown (dual-sgRNA median knockdown 82%; Dolcetto median knockdown 65%; Mann-Whitney p-value = $2.4 \cdot 10^{-7}$) as well as the strongest knockdown per gene (dual-sgRNA median knockdown 90%; Dolcetto median knockdown 87%; Mann-Whitney p-value = $2 \cdot 10^{-4}$; *Figure 1F*). Indeed, the top-ranked element of our dual-sgRNA library (guides 1+2) alone produced comparable knockdown to the best of all three Dolcetto sgRNAs (dual-sgRNA element 1+2 median knockdown 86%; best Dolcetto sgRNA median knockdown 87%; Mann-Whitney p-value = 0.43) (*Figure 1G*). We note that an analogous dual-sgRNA approach may improve knockdown for the Dolcetto library. Nonetheless, from these data we conclude that our dual-sgRNA library improves on-target knockdown compared to gold-standard CRISPRi libraries.

## Design of CRISPRi effector expression constructs for systematic comparisons

We next sought to compare different CRISPRi effectors, with the goal of identifying an effector with strong activity and minimal non-specific effects on global transcription and cell growth. We selected four repressor domains that had been described to mediate strong and specific knockdown in dCas9 fusions: (1) the KRAB domain from KOX1 (*ZNF10*), which was used in the original conception of CRISPRi for mammalian cells *Gilbert et al., 2013*; (2) the KRAB domain from ZIM3, which was recently reported to mediate stronger knockdown than KRAB(KOX1) *Alerasool et al., 2020*; (3) the SIN3A interacting domain of MAD1 (SID4x) *Carleton et al., 2017*; and (4) the transcription repression domain of MeCP2 (*Yeo et al., 2018*).

To enable direct comparisons, we embedded each effector in a standardized lentiviral expression construct (*Figure 2A*, *Supplementary file 7*). Briefly, in this construct, expression is driven by a spleen focus-forming virus (SFFV) promoter, with an upstream ubiquitous chromatin opening element (UCOE) to minimize silencing, internal nuclear localization signals (NLSs), and an internal HA tag, a GFP marker linked at the C-terminus via a P2A ribosomal skipping sequence to allow for stable cell line generation by fluorescence-activated cell sorting (FACS), and a woodchuck hepatitis virus post-transcriptional regulatory element in the 3′ UTR to increase mRNA stability. Where necessary, we included linker sequences derived from the XTEN domain (*Schellenberger et al., 2009*), which is reported to be functionally innocuous, to minimize proteolytic cleavage between dCas9 and fused repressor domains. We attempted to maximize the activity for each repressor domain based on our previous data and data in the literature, although we note that our evaluation is not exhaustive. The final designs of the four effector expression constructs are depicted in *Figure 2—figure supplement 1*, with further rationale in the Materials and methods section. We then compared the four effectors with regards to two key criteria: on-target activity and absence of non-specific effects on cell viability and gene expression.

## CRISPRi effectors containing SID or MeCP2 domains have non-specific effects on cell viability and gene expression

The repressor domain of each CRISPRi effector is a transcription factor domain whose overexpression has the potential to cause non-specific (i.e., not mediated by dCas9 targeting) and potentially detrimental effects on transcription or cell proliferation. To test for effects on proliferation, we generated K562 cell lines stably expressing each effector by lentiviral transduction followed by FACS (*Figure 2B*) and then quantified the effect of each effector on cell proliferation using an internally normalized competitive growth assay. We mixed cells bearing each effector ~1:1 with cells expressing mCherry and quantified growth defects of effector-expressing cells by measuring the ratio of mCherry-negative to mCherry-positive cells over time by flow cytometry. We used mCherry-expressing cells as a reference population instead of parental, GFP-negative cells because some of the effector-expressing cells convert to GFP-negative over time due to silencing, which is difficult to separate from true dropout of effector-expressing cells due to growth defects. Over 19 days, cells expressing dCas9 only, dCas9-Kox1, or Zim3-dCas9 proliferated at the same rate as cells expressing GFP only or non-transduced control cells, suggesting that expression of these effectors is not toxic over this time span (*Figure 2C*). By contrast, cells expressing SID-dCas9-Kox1 had a strong growth defect (~6% per day), and cells expressing dCas9-Kox1-MeCP2 had a mild growth defect (~1% per day, *Figure 2C*).

To assess non-specific effects of effectors on transcription, we performed global transcriptome profiling of K562 cells stably transduced with these effectors by RNA-seq (*Figure 2D and E*). Consistent with the growth assay, cells expressing SID-dCas9-Kox1 had globally perturbed transcription, with 4282 genes differentially expressed compared to control cells expressing GFP only at $p < 0.05$ (*Figure 2E*). Indeed, these samples clustered separately from every other control and effector-expressing sample (*Figure 2D*). In addition, 53 genes were differentially expressed in cells with dCas9-Kox1-MeCP2, suggesting that constitutive expression of this effector also leads to minor non-specific effects on transcription (*Figure 2E*). No more than three genes were detected to be differentially expressed in cells expressing any of the other effectors, suggesting that these effectors do not non-specifically perturb transcription (*Figure 2E*). Together, these results suggest that (over)expression of SID-dCas9-Kox1 is toxic and globally perturbs transcription at least in K562 cells. We therefore excluded this effector from further analysis.

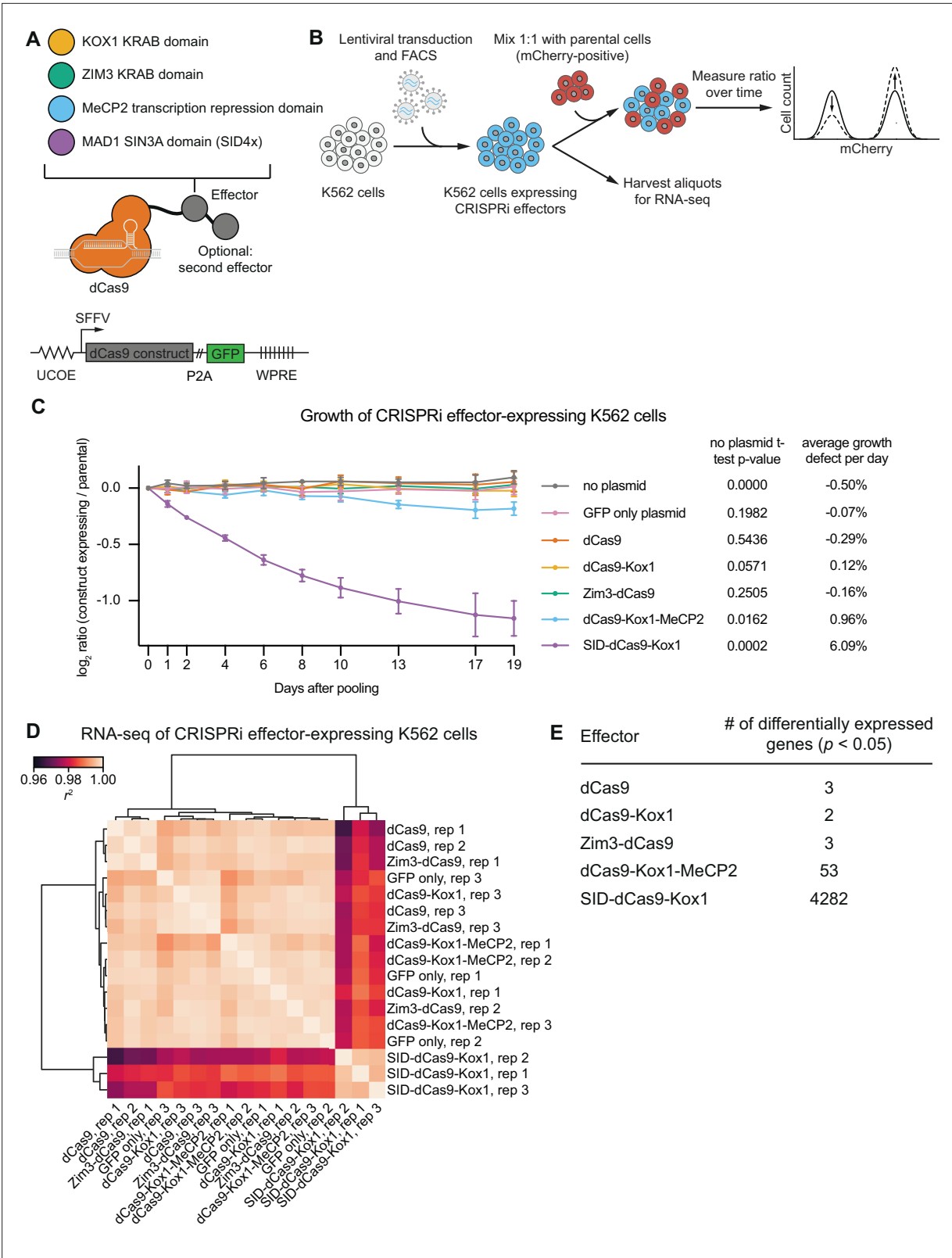

**Figure 2.** CRISPR interference (CRISPRi) effectors containing SID or MeCP2 domains have non-specific effects on cell viability and gene expression. (**A**) Schematics of CRISPRi transcription repressor domains and general lentiviral expression construct used for all CRISPRi effectors. UCOE = ubiquitous chromatin opening element; SFFV = spleen focus-forming virus promoter; P2A = ribosomal skipping sequence; WPRE = woodchuck hepatitis virus post-transcriptional regulatory element. Further information on repressor domains and lentiviral expression constructs can be found in the main text

*Figure 2 continued on next page*

Figure 2 continued

and Materials and methods. (**B**) Experimental design to test effects of stable expression of each CRISPRi effector on growth and transcription in K562 cells. (**C**) Growth defects of effector-expressing cells, measured as the $\log_2$ of the ratio of mCherry-negative (effector-expressing) to mCherry-positive (not effector-expressing) cells in each well normalized to the same ratio on day 0. mCherry levels were measured for 19 days after pooling cells. Data represent mean ± SD from three independent transductions of expression constructs. p-Values are from an unpaired two-tailed t-test comparing D19 values for each sample to the D19 value for the 'no plasmid' sample. Average percent growth defect per day is the $\log_2$ D19 value divided by the number of days, multiplied by 100 for a percent value. (**D**) Clustered heatmap of correlation of transcript counts from K562 cells expressing indicated CRISPRi effectors or a GFP control. Correlations across samples were calculated using normalized counts (reads per million) for all genes with mean normalized count >1 and then clustered using the Ward variance minimization algorithm implemented in scipy. $r^2$ is squared Pearson correlation. Data represent three independent transductions of expression constructs. (**E**) Number of differentially expressed genes ($p<0.05$) for cells expressing each effector versus cells expressing GFP only. p-Values were calculated using a Wald test and corrected for multiple hypothesis testing as implemented in DeSeq2.

The online version of this article includes the following source data and figure supplement(s) for figure 2:

**Source data 1.** p-Values and growth defects depicted in *Figure 2C*.

**Source data 2.** Data depicted in *Figure 2E*.

**Figure supplement 1.** Design of constructs for CRISPR interference (CRISPRi) effector expression.

## Zim3-dCas9 and dCas9-Kox1-MeCP2 mediate strongest knockdown

We next sought to measure the efficacy of each effector in knocking down targeted genes with two complementary approaches: (i) measurement of growth phenotypes resulting from knockdown of essential genes, that is, genes required for the growth or survival of dividing human cells, and (ii) direct measurement of knockdown of cell surface proteins (*Figure 3A*, *Supplementary file 8*). In both assays, we used single-sgRNA expression cassettes, which allowed us to use previously validated strong and intermediate-activity sgRNAs (*Jost et al., 2020*). We included intermediate-activity sgRNAs for two reasons: First, activity differences between effectors are more apparent when knockdown is not saturated. Second, as it can be challenging to identify sgRNAs with high activity across genes and cell types, effectors that mediate strong knockdown even with imperfect sgRNAs could reduce false-negative rates in genetic screens.

We measured growth phenotypes resulting from knockdown of essential genes using internally normalized competitive growth assays. We transduced K562 cell lines stably expressing each CRISPRi effector with vectors simultaneously expressing an sgRNA and a fluorescent marker (mCherry) at a low multiplicity of infection (0.2–0.5). We then monitored the ratio of sgRNA-expressing cells (mCherry+) and unperturbed cells (mCherry-) by flow cytometry, with the expectation that cells with an essential gene-targeting sgRNA would deplete at a rate proportional to CRISPRi activity. We targeted three genes, alanyl-tRNA synthetase (*AARS*), the mitochondrial inner membrane import factor DNAJC19, and subunit D of RNA polymerase I and III (*POLR1D*), with three different sgRNAs each. For all sgRNAs tested, sgRNA-expressing cells depleted at the fastest rate with Zim3-dCas9 and at the second-fastest rate with either dCas9-Kox1 or dCas9-Kox1-MeCP2 (*Figure 3B*, *Figure 3—figure supplement 1A*).

Next, to directly measure depletion of targeted proteins, we knocked down the non-essential cell surface proteins CD55 (complement decay-accelerating factor), CD81 (TAPA-1/TSPAN28), and CD151 (TSPAN24) and measured staining intensity with fluorescently labeled antibodies by flow cytometry as a proxy for protein levels. We transduced K562 lines stably expressing the different CRISPRi effectors with vectors expressing either targeting or non-targeting sgRNAs at a low multiplicity of infection (0.2–1). Six days after transduction, we stained cells with fluorescently labeled antibodies against the different cell surface proteins and assessed knockdown by comparing the median antibody staining intensity in cells expressing a targeting sgRNA and cells expressing a non-targeting control sgRNA. With strong sgRNAs, Zim3-dCas9, dCas9-Kox1, and dCas9-Kox1-MeCP2 all mediated strong depletion of each cell surface protein (>96.8% median depletion for all effectors and sgRNAs). With weak sgRNAs, dCas9-Kox1-MeCP2 mediated the strongest knockdown closely followed by Zim3-dCas9, whereas dCas9-Kox1 mediated weaker knockdown (*Figure 3C and D*, *Figure 3—figure supplement 1B–D*).

Importantly, flow cytometry reports on expression at the single-cell level, allowing us to assess cell-to-cell heterogeneity in knockdown, which is missed when quantifying median expression. As a proxy for heterogeneity, we calculated the fraction of cells without evidence of knockdown despite the use of a strong sgRNA (*Figure 3E*). For Zim3-dCas9, knockdown was largely homogeneous, with

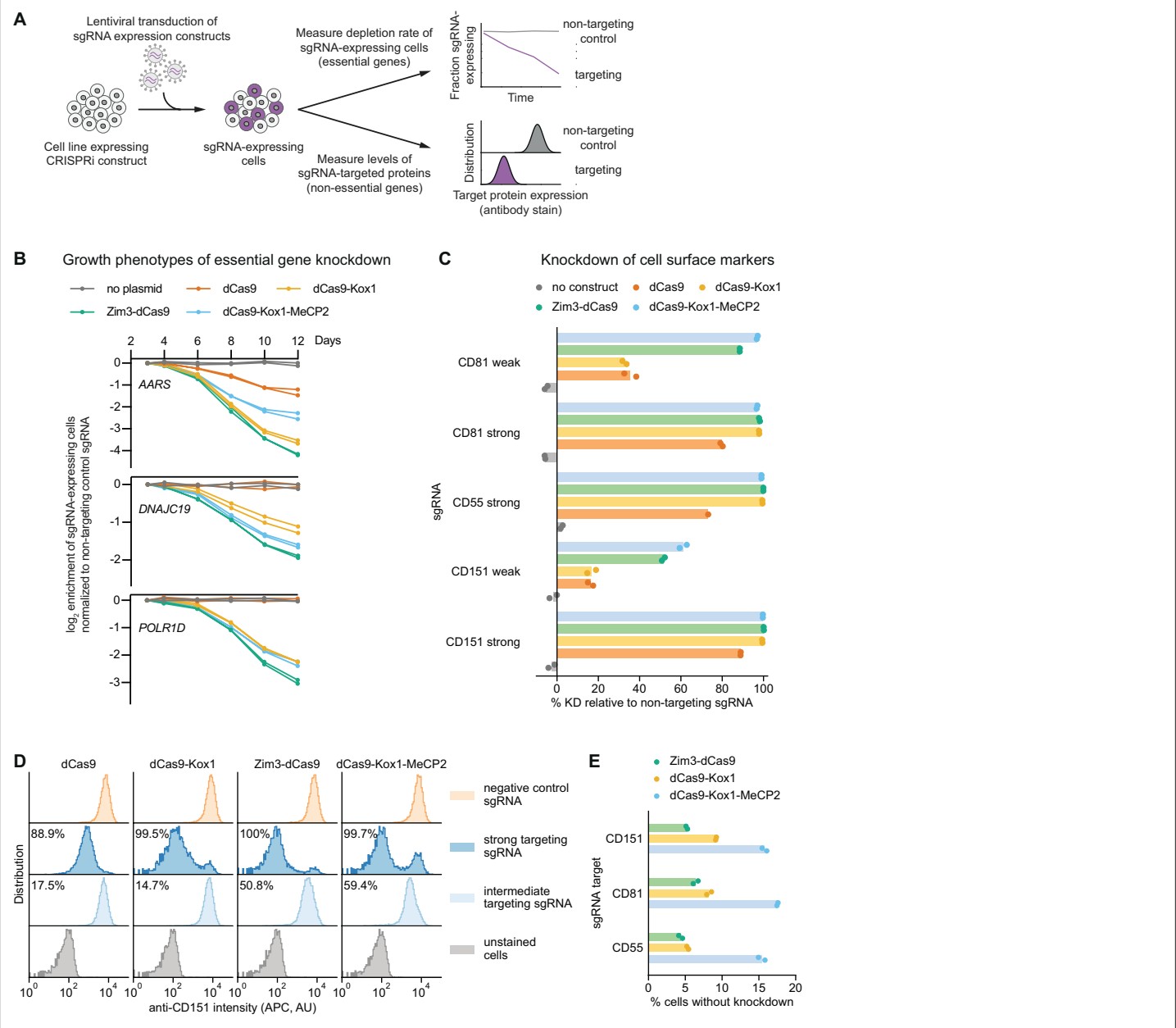

**Figure 3.** Zim3-dCas9 and dCas9-Kox1-MeCP2 mediate strongest knockdown. (**A**) Experimental design to measure knockdown mediated by different CRISPR interference (CRISPRi) effectors by delivering single guide RNAs (sgRNAs) targeting either essential genes or cell surface markers. (**B**) Depletion of K562 cells expressing essential gene-targeting sgRNAs and different CRISPRi effectors, measured as the ratio of mCherry-positive (sgRNA-expressing) to mCherry-negative (not sgRNA-expressing) cells in a given well. mCherry levels were measured for 12 days after transduction, starting on day 3. Data from two replicate transductions. (**C**) Percent knockdown of cell surface markers by different CRISPRi effectors in K562 cells. Cell surface marker levels were measured on day 6 post-transduction by staining with an APC-conjugated antibody. Knockdown was calculated as the ratio of median APC signal in sgRNA-expressing cells and median APC signal in cells expressing a non-targeting control sgRNA after subtraction of background APC signal. Data from two replicate transductions. Cells expressing dCas9 and a strong CD55-targeting sgRNA are represented by a single replicate. (**D**) Distribution of anti-CD151 signal intensity (APC) in individual cells from one representative transduction. Data from second replicate are shown in *Figure 3—figure supplement 1B*. Knockdown was quantified as in **C** as the ratio of the median APC signals. (**E**) Percentage of cells without observable knockdown despite expressing a strong sgRNA, as quantified from the fluorescence distributions.

The online version of this article includes the following figure supplement(s) for figure 3:

**Figure supplement 1.** Additional measurements of on-target activity of CRISPR interference (CRISPRi) effectors.

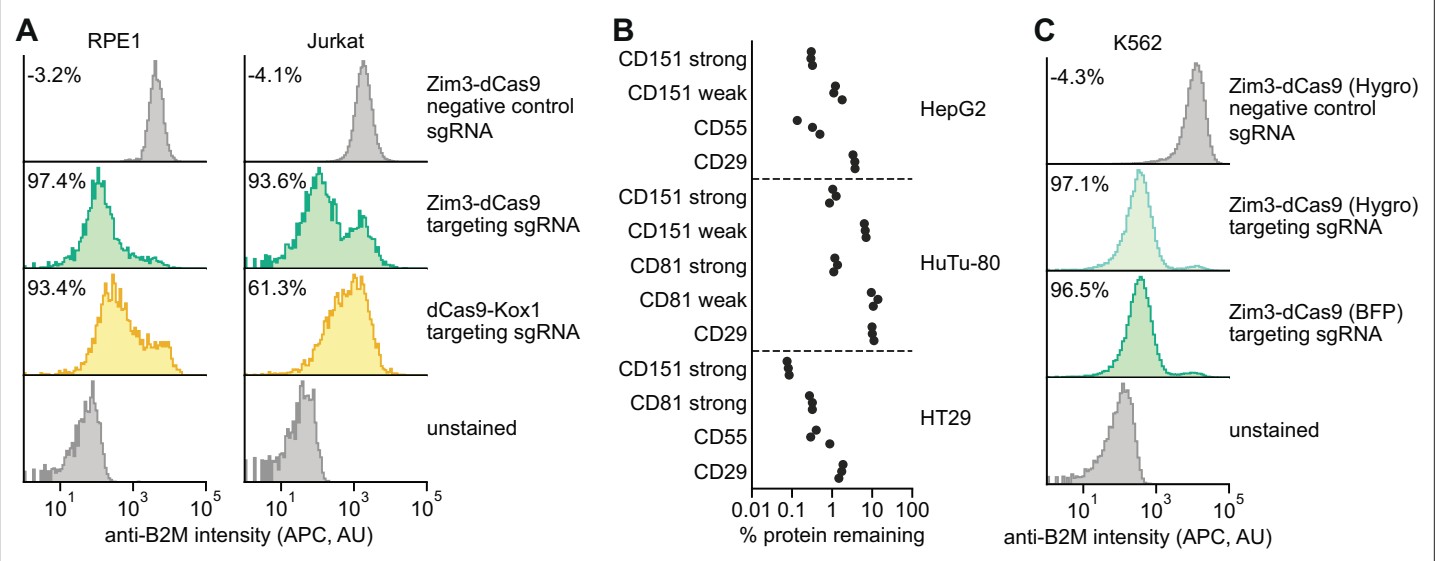

**Figure 4.** Validation of a suite of optimized Zim3-dCas9 cell lines. (**A**) Distribution of anti-B2M signal intensity (APC) in individual RPE1 (left) and Jurkat (right) cells expressing indicated CRISPR interference (CRISPRi) effectors and single guide RNAs (sgRNAs). Knockdown was calculated as the ratio of median APC signal in transduced (sgRNA-expressing) cells and median APC signal in non-transduced cells in the same well, after subtraction of background APC signal. (**B**) Depletion of indicated cell surface markers in HepG2 (top), HuTu-80 (middle), and HT29 (bottom) cells expressing Zim3-dCas9. Cell surface marker levels were measured 6–14 days post-transduction by staining with APC-conjugated antibodies. Knockdown was calculated as the ratio of median APC signal in sgRNA-expressing cells and median APC signal in cells expressing a non-targeting control sgRNA after subtraction of background APC signal. (**C**) Distribution of anti-B2M signal intensity (APC) in individual K562 cells expressing indicated CRISPRi effectors and sgRNAs. The Zim3-dCas9 (Hygro) cell line was generated by transduction followed by hygromycin selection and does not express a fluorescent protein. Knockdown was calculated as in **A**.

The online version of this article includes the following figure supplement(s) for figure 4:

**Figure supplement 1.** Single-cell distributions of knockdown in different Zim3-dCas9 cell lines.

**Figure supplement 2.** Growth of different Zim3-dCas9-expressing cell lines.

only ~5% of cells without detectable knockdown (***Figure 3D and E***, ***Figure 3—figure supplement 1B–D***), perhaps due to the presence of some senescent cells in the population in which lack of cell division limits protein dilution. By contrast, for dCas9-Kox1-MeCP2, 15–20% of cells did not achieve knockdown (***Figure 3D and E***, ***Figure 3—figure supplement 1B–D***). This result may be explained by the toxicity of the effector protein leading to selection against effector expression (***Figure 2C***) or by larger cell-to-cell variability in expression levels of the dCas9-Kox1-MeCP2 effector or may be indicative of an intrinsic property of MeCP2 activity. The observed heterogeneity in MeCP2 knockdown may help explain why dCas9-Kox1-MeCP2 appears to mediate the strongest median knockdown while Zim3-dCas9 leads to faster dropout of sgRNA-expressing cells in the essential gene growth assay; in the growth assay, heterogeneity would lead to worse performance due to selection against strong knockdown. In sum, these results suggest that the Zim3-dCas9 effector confers strong knockdown that is homogeneous across a cell population.

## A versatile collection of Zim3-dCas9 constructs for robust knockdown across cell types

To assess the general utility of the Zim3-dCas9 effector, we measured knockdown efficacy in different cell types. For each cell type, we constructed cell lines stably expressing Zim3-dCas9 (see Materials and methods) and measured knockdown of cell surface proteins by flow cytometry. In both RPE1 (retinal pigment epithelium) and Jurkat (acute T-cell leukemia) cells, cells expressing Zim3-dCas9 had stronger knockdown than previously reported cell lines expressing dCas9-Kox1 (***Figure 4A***; ***Horlbeck et al., 2018***; ***Jost et al., 2017***). Zim3-dCas9 also conferred strong and homogeneous knockdown in HepG2 (hepatocellular carcinoma), HT29 (colorectal adenocarcinoma), and HuTu-80 (duodenal

adenocarcinoma) cells (*Figure 4B*, *Figure 4—figure supplement 1*) without negative impacts on cell proliferation (*Figure 4—figure supplement 2*).

To further maximize utility of the Zim3-dCas9 effector, we generated a panel of constructs for expression of Zim3-dCas9 from the SFFV or EF1α promoters linked to BFP, GFP, or mCherry. We also generated backbones to express effectors from additional promoters (CMV, EFS) and with different types of C-terminal fluorescent protein linkages (P2A, internal ribosome entry site [IRES], direct fusion). In addition, as the bright fluorescence from the fluorescent proteins may be undesirable in some settings, we generated a construct in which expression of Zim3-dCas9 is linked to a hygromycin resistance marker (Zim3-dCas9 [Hygro]). K562 cells stably transduced with Zim3-dCas9 (Hygro) and selected with hygromycin for 4 weeks had strong and homogeneous knockdown that was indistinguishable from knockdown in a cell line generated by FACS (*Figure 4C*). Finally, we generated constructs in which the fluorescent protein is flanked by LoxP sites, such that the fluorescent protein can be removed by transient delivery of Cre once a stable cell line has been generated. A full list of our constructs is included in *Supplementary file 7*. All constructs are available via Addgene. Finally, to help readers identify CRISPRi constructs appropriate for their needs and generate CRISPRi cell lines, we generated a comprehensive CRISPRi cell line generation protocol, which is available at https://www.jostlab.org/resources/ and https://weissman.wi.mit.edu/resources/. Our collection of Zim3-dCas9 expression constructs and streamlined protocols enables robust CRISPRi across a broad range of cell models.

## Discussion

High-quality genetic screening approaches are catalysts for basic research and drug development. Among the available approaches, CRISPRi has several appealing features including independence of double-stranded DNA breaks, homogeneity and reversibility of perturbations, accessibility of partial loss-of-function phenotypes, and compatibility with direct measurements of target gene expression levels in both bulk populations and single cells. CRISPRi screens have indeed propelled biological discovery in several contexts, but broader deployment has been limited by difficulties in generating CRISPRi cell models and limited knockdown efficacy for a subset of genes. Here, we present a suite of tools and accompanying protocols to address these limitations and improve the efficacy and accessibility of CRISPRi.

Our ultra-compact, dual-sgRNA CRISPRi library confers stronger knockdown and growth phenotypes than previously reported libraries and thus should minimize false-negative rates in screens. Nonetheless, this library also has drawbacks. First, some library elements undergo intermolecular recombination during lentiviral transduction. We can detect and computationally remove such recombination events, such that they do not corrupt the resulting data. As a consequence, recombination primarily decreases effective library coverage, and in return cell numbers need to be increased by ~20–30% to ensure coverage. In the future, recombination may be further mitigated using decoy vectors, different promoters, and alternatively processed guides (*Adamson et al., 2016*; *Dong et al., 2017*; *Feldman et al., 2018*; *Knapp et al., 2019*). Second and perhaps more importantly, screens will be inherently noisier and sensitive to off-target effects with only a single element per gene, such that in standard cell lines in which cell numbers are not a concern, existing single-sgRNA libraries may remain the approach of choice. Inclusion of the 3–4 and 5–6 element sublibraries in our dual-sgRNA library can mitigate this noise at the expense of some of the compactness. In cases in which cell numbers are limited by the model, time, or cost, however, the compactness of our dual-sgRNA library will be transformative. Examples include screens in primary or stem cell-derived models or in vivo as well as screens with high-content readout such as Perturb-seq (*Bock et al., 2022*; *Przybyla and Gilbert, 2022*). In sum, our dual-sgRNA libraries improve CRISPRi knockdown and complement existing libraries by broadening the scope of models in which CRISPRi screens are feasible.

The dual-sgRNA strategy may provide similar benefits for other CRISPR modalities such as CRISPR-mediated overexpression (CRISPR activation [CRISPRa]), as also described by others (*Yin et al., 2022*), and in mouse cells. We have cloned a human dual-sgRNA CRISPRa library (*Supplementary file 9*) and designed in silico dual-sgRNA CRISPRi and CRISPRa libraries targeting the mouse genome (*Supplementary files 10 and 11*). Because few human CRISPRa and mouse screens have been reported, we did not use empirical sgRNA selection for these libraries and instead ranked sgRNAs by their predicted activities (*Horlbeck et al., 2016a*). Finally, the improved knockdown afforded by the dual-sgRNA

approach will also be beneficial in arrayed experiments, in which recombination is not a concern, and we have generated a protocol for cloning dual-sgRNA libraries in array (available at https://www.jostlab.org/resources/ and https://weissman.wi.mit.edu/resources/).

In the realm of CRISPRi effectors, our work points to a clear consensus: Zim3-dCas9 is the effector of choice, as it appears equal or superior to other effectors in every test we performed and had no apparent downsides. We had previously measured by Perturb-seq that Zim3-dCas9 afforded median mRNA knockdown of 91.6% across 2285 genes in RPE1 cells (*Replogle et al., 2022*), and here we further found that Zim3-dCas9 mediates robust knockdown across a range of cell types. Our work highlights the importance of using multiple assays to assess effector activity including single-cell assays to assess cell-to-cell heterogeneity, of directly measuring knockdown instead of relying on proxies such as growth phenotypes that conflate multiple factors, and of evaluating effectors in stably transduced cells rather than in transiently transfected cells to evaluate longer-term consequences for cell viability. To facilitate implementation of CRISPRi in additional cell models, we created a suite of effector expression constructs with different combinations of promoters and markers (*Supplementary file 7*) as well as a cell line generation protocol. After generating a new CRISPRi cell line, we recommend evaluating the cell line for activity and lack of non-specific effects on cell growth, using for example the protocols and assays we describe, before proceeding to large-scale experiments.

Nonetheless, there is more progress to be made in evaluating effectors and generating robust CRISPRi models. First, our comparison of the effectors was not exhaustive. For example, although we expressed all effectors from the same context, we did not control for potential differences in expression levels or nuclear localization across effectors. The Zim3-dCas9 expression constructs appear optimal as they are, but activities of other effectors may be boosted by optimizing these factors. Second, repression of gene expression is generally mediated through recruitment of endogenous cofactors; for KRAB domains such as those from Zim3 and Kox1, this endogenous cofactor is TRIM28 (*Ecco et al., 2017*). TRIM28 expression varies by cell type, and efficacy of Zim3 and Kox1 may be limited in cell types with low TRIM28 expression. In such cell types, the MeCP2 effector may be a suitable alternative, but the selection against effector-expressing cells may increase false-positive and false-negative rates. Third, we did not measure if the effectors differed in propensity for sgRNA-dependent off-target effects. Previous work on dCas9-Kox1 had documented that well-designed sgRNAs have minimal off-target effects (*Gilbert et al., 2013*). The main source of off-target effects of CRISPRi are at bidirectional promoters, which likely is an inevitable consequence of the mechanism of CRISPRi. We note that the stronger activity of Zim3-based effectors may amplify such effects. For now, such off-target effects can be readily predicted and measured, for example by Perturb-seq (*Replogle et al., 2022*). Perhaps future efforts will identify strategies to limit knockdown of neighboring genes. Finally, in some cell types effector expression is silenced over time, leading to loss of CRISPRi activity. We described some strategies to counteract such silencing in our cell line generation protocol, but further protection against silencing remains as an area for improvement. In any case, the assays we describe may be used to test additional effectors in a streamlined and standardized fashion, with the goal of making CRISPRi universally available across cell models.

Altogether, our resources and best practices will guide both current implementations and future developments of CRISPRi. All our protocols, constructs, cell lines, and libraries are available as resources to the community.

## Materials availability

All sgRNA expression plasmids, sgRNA libraries, and effector expression plasmids are available via Addgene, with accession numbers listed in *Supplementary file 7*. All new CRISPRi cell lines are available from the corresponding authors. Detailed protocols for design and cloning of pooled and arrayed dual-sgRNA libraries, sequencing of samples from screens with dual-sgRNA libraries, and generation of cell lines with robust CRISPRi activity are publicly available at https://www.jostlab.org/resources/ and https://weissman.wi.mit.edu/resources/.

## Data and code availability

Python scripts for alignment of sequencing data from dual-sgRNA screens with and without IBCs are available here: https://github.com/josephreplogle/CRISPRi-dual-sgRNA-screens. Sequencing data are

available on NCBI GEO under accession number GSE205310 (Perturb-seq) and GSE205147 (bulk RNA-seq).

# Materials and methods

## Key resources table

A key resources table is included as Appendix 1.

### Cell line generation and maintenance

K562 cells were grown in RPMI 1640 medium with 25 mM HEPES, 2 mM L-glutamine, 2 g/L NaHCO$_3$ (Gibco) supplemented with 10% (v/v) standard fetal bovine serum (FBS, VWR), 100 units/mL penicillin, 100 µg/mL streptomycin, and 2 mM L-glutamine (Gibco). hTERT-immortalized RPE1 cells (ATCC CRL-4000) were grown in Dulbecco's modified eagle medium (DMEM):F12 (Gibco) supplemented with 10% (v/v) standard FBS (VWR), 0.01 mg/mL hygromycin B, 100 units/mL penicillin, and 100 µg/mL streptomycin. Jurkat cells (Clone E6-1, ATCC TIB-152) were grown in RPMI 1640 medium with 25 mM HEPES, 2 mM L-glutamine, 2 g/L NaHCO$_3$ (Gibco) supplemented with 10% (v/v) standard FBS (VWR), 100 units/mL penicillin, 100 µg/mL streptomycin, and 2 mM L-glutamine (Gibco). HepG2 (ATCC HB-8065) and HuTu-80 cells (ATCC HTB-40) were grown in Eagle's minimum essential medium with 1.5 g/L NaHCO$_3$, 110 mg/L sodium pyruvate, 292 mg/L L-glutamine (Corning) supplemented with 10% (v/v) standard FBS (R&D Systems), 100 units/mL penicillin, and 100 µg/mL streptomycin (Gibco). HT29 cells (ATCC HTB-38) were grown in DMEM with 25 mM D-glucose, 3.7 g/L NaHCO$_3$, 4 mM L-glutamine (Gibco) supplemented with 10% (v/v) standard FBS (R&D Systems), 100 units/mL penicillin, and 100 µg/mL streptomycin (Gibco). HEK293T cells were grown in DMEM with 25 mM D-glucose, 3.7 g/L NaHCO$_3$, 4 mM L-glutamine (Gibco), and supplemented with 10% (v/v) standard FBS (VWR or R&D Systems), 100 units/mL penicillin, 100 µg/mL streptomycin, and 2 mM L-glutamine (Gibco). K562 (chronic myelogenous leukemia) and HT29 (colorectal adenocarcinoma) cells are derived from female patients. Jurkat (acute T-cell leukemia), HuTu-80 (duodenal adenocarcinoma), and HepG2 (hepatocellular carcinoma) cells are derived from male patients. HEK293T (embryonic kidney) cells are derived from a female fetus. RPE1 (immortalized retinal pigment epithelium) cells are derived from a female subject. All cell lines were grown at 37°C in the presence of 5% CO$_2$. RPE1, Jurkat, HepG2, HuTu-80, and HT29 cells were obtained as fresh stocks from ATCC for this work and not authenticated further. All cell lines were tested regularly (approximately every 6 months) for *Mycoplasma* using either the PCR-based Universal Mycoplasma Detection Kit (ATCC) or the luminescence-based MycoAlert Kit (Lonza).

To generate the K562 cell lines stably expressing various CRISPRi effectors, parental K562 cells were stably transduced with lentiviral vectors expressing the corresponding effectors linked to GFP via a P2A ribosome skipping sequence from an SFFV promoter with an upstream UCOE. Polyclonal populations of GFP-positive cells were selected using two rounds of FACS on a Sony SH800S Cell Sorter.

To generate RPE1 cells stably expressing Zim3-dCas9, RPE-1 cells were infected with lentivirus containing UCOE-SFFV-Zim3-dCas9-P2A-BFP (pJB108) at low multiplicity of infection by centrifugation at 1000 × *g*. Polyclonal populations of BFP-positive cells were selected using two rounds of FACS on a Sony SH800S Cell Sorter. To generate Jurkat cells stably expressing Zim3-dCas9, Jurkat cells were infected with virus containing UCOE-EF1α-Zim3-dCas9-P2A-mCh (pJB109) at low multiplicity of infection by centrifugation at 1000 × *g*. Polyclonal populations of mCherry-positive cells were selected using two rounds of FACS on a Sony SH800S Cell Sorter. To generate HepG2, HuTu-80, and HT29 cells stably expressing Zim3-dCas9, cells were infected with lentivirus containing UCOE-EF1α-Zim3-dCas9-P2A-mCh (pJB109) at low multiplicity of infection. Polyclonal populations of mCherry-positive cells were selected using two rounds of FACS on a FACSAria II Cell Sorter (BD Biosciences). To generate K562 cells stably expressing Zim3-dCas9 without a fluorescent marker, K562 cells were infected with virus containing UCOE-SFFV-Zim3-dCas9-P2A-hygro (pAG389) at low multiplicity of infection by centrifugation at 1000 × *g*. To select for a polyclonal population, cells were treated 48 hr after infection with 200 µg/mL hygromycin for 1 week, followed by treatment 500 µg/mL hygromycin for 3 weeks.

## Lentivirus production

Lentivirus was generated by transfecting HEK239T cells with the transfer plasmid and four packaging plasmids (for expression of VSV-G, Gag/Pol, Rev, and Tat) using TransIT-LT1 Transfection Reagent (Mirus Bio). Viral supernatant was harvested 2 days after transfection and filtered through 0.44 µm PES filters and/or frozen at –80°C prior to transduction.

## Design and cloning of pilot genome-wide single- and dual-sgRNA CRISPRi libraries

To compare the use of single- and dual-sgRNA CRISPRi libraries in systematic genetic screens, pilot genome-wide single- and dual-sgRNA CRISPRi libraries were designed and cloned. sgRNAs targeting each gene were selected from our previously published hCRISPRi v2 library by balancing empirical data from previous genetic screens with Horlbeck et al. predicted rankings (*Horlbeck et al., 2016a*) using a three-tiered approach:

Tier 1: For genes essential for growth in the K562 CRISPRi screen data (p-value <0.001 and γ<–0.2) (*Horlbeck et al., 2016a*), sgRNAs were ranked by their growth phenotype.

Tier 2: As many genetic perturbations only cause a conditional cellular phenotype (e.g., in a particular cell type, chemical stressor, or reporter phenotype), we next aggregated data across multiple genetic screens (only a subset of the data in *Supplementary file 3* was available for the pilot library design). For genes that were identified as a significant hit (FDR 0.05 based on MAGeCK RRA p-value; *Li et al., 2014*) in previous CRISPRi screens, sgRNAs were ranked by the sum of *z*-scored phenotypes across screens.

Tier 3: For all other genes, sgRNAs were ranked by the regression scores in hCRISPRi v2.1 (*Horlbeck et al., 2016a*).

Using this ranking scheme, we selected the single best sgRNA for a single-sgRNA/single-element-per-gene library (dJR004) and the two best sgRNAs for a dual-sgRNA/single-element-per-gene library (dJR020). A list of sgRNA targeting sequences both the single- and dual-sgRNA libraries is available in *Supplementary file 1*.

The single-sgRNA library dJR004 was cloned using the protocol described here: https://weissman. wi.mit.edu/resources/Pooled_CRISPR_Library_Cloning.pdf. A modified CROP-seq sgRNA lentiviral expression vector (pJR107) was derived from the parental vector pBA950 (https://www.addgene.org/ 122239/) by incorporating a GFP fluorescent marker and a UCOE element upstream of the EF1alpha promoter to prevent marker silencing. sgRNA targeting sequences were appended with flanking sequence, BstX1/BlpI overhangs, and PCR adapters. The library was synthesized as an oligonucleotide pool (Twist Biosciences), PCR-amplified, BstX1/BlpI-digested, and inserted into pJR107 by ligation.

The dual-sgRNA library dJR020 was cloned using the protocol available at https://www.jostlab. org/resources/ and https://weissman.wi.mit.edu/resources/. Briefly, dual-sgRNA targeting sequences were spaced by a BsmBI cut site and appended with flanking sequence, BstX1/BlpI overhangs, and PCR adapters with the structure: with the structure: 5'- PCR adaptor – CCACCTTGTTG – targeting sequence A – gtttcagagcgagacgtgcctgcaggatacgtctcagaaacatg – targeting sequence B – GTTTAAGA GCTAAGCTG – PCR adaptor-3'. The library was synthesized as an oligonucleotide pool (Twist Biosciences), PCR-amplified, BstX1/BlpI-digested, and inserted into pJR104 (Addgene #187243) by ligation. Next, the sgRNA CR3/hU6 promoter insert pJR98 (Addgene #187239) was BsmBI-digested and ligated into the BsmBI-digested library to generate the final library. In the final library, each element expresses two unique sgRNAs from tandem U6 expression cassettes.

## Genome-wide CRISPRi growth screens in K562 cells for library comparison

Parallel growth screens were performed to compare dJR004 versus dJR020. Lentivirus from dJR004 and dJR020 was produced in HEK293T as described above. CRISPRi K562 cells expressing dCas9-Kox1 were spinfected (1000G) with polybrene (8 µg/mL, Sigma-Aldrich) with lentivirus from dJR004 and dJR020 in biological replicate. Throughout the screen, cells were maintained at a density between 250,000 and 1,000,000 cells/mL and 1000× coverage per library element. On day 3 post-transduction, an infection rate of 11–18% was measured by GFP fluorescence. On day 3 through day 6 post-transduction, puromycin at 1 µg/mL was used to select for infected cells, and cells were allowed to recover for 2 days. On day 8 post-transduction, a cell pellet was frozen for each replicate

representing the initial sample ($T_0$) of the screen. Throughout the screen, the number of cell doublings was recorded, and final samples ($T_{final}$) were collected on day 20 post-transduction.

## Assessment of recombination rates in RPE1 and Jurkat cells

We cloned a dual-sgRNA library targeting DepMap Common Essential genes (n=2291 dual-sgRNA elements, *Supplementary file 12*) using the protocol available at https://www.jostlab.org/resources/ and https://weissman.wi.mit.edu/resources/. We then transduced pools of RPE1 and Jurkat cells with this library at 1000× coverage per library element, harvested genomic DNA from cells at day 7 post-transduction, amplified sgRNA cassettes from extracted genomic DNA, and sequenced as outlined above to quantify sgRNA recombination rates in different cell types.

## Screen library preparation, sequencing, and analysis

Amplicon DNA libraries were prepared from cell pellets as previously described (*Nuñez et al., 2021*). Genomic DNA was isolated using a NucleoSpin Blood XL kit or NucleoSpin Blood L kit (Macherey-Nagel) depending on pellet size. Purified genomic DNA was directly amplified by 22 cycles of PCR using NEBNext Ultra II Q5 PCR MasterMix (NEB). Sequencing was performed on a NovaSeq 6000 (Illumina) using a 19 bp Read 1, 19 bp Read 2, and 5 bp Index Read 1 with custom sequencing primers.

After sequencing, sgRNA sequencing reads were aligned to the single- and dual-sgRNA libraries using a custom Python script without allowing mismatches. Reads for which the Read 1 and Read 2 sgRNA sequences did not target the same gene likely arose from lentiviral recombination and were discarded from downstream analysis. For both replicates of the dual-sgRNA library, 29.4% of mapped reads contained sgRNAs targeting different genes. Library elements (i.e., sgRNAs or sgRNA pairs) represented with 0 sequencing reads were assigned a pseudocount of 1 read, while library elements represented with fewer than 50 sequencing reads in both $T_0$ and $T_{final}$ of any screen replicate were excluded from analysis. For each sgRNA or sgRNA pair, the growth phenotype ($\gamma$) was defined as the $\log_2$(sgRNA normalized count $T_{final}$/sgRNA normalized $T_0$) – median non-targeting control $\log_2$(sgRNA normalized count $T_{final}$/sgRNA normalized count $T_0$), divided by the replicate total cell doublings and normalized to the total number of sequencing reads for a given sample (*Gilbert et al., 2014*). Read counts and growth phenotypes of library elements are included in *Supplementary file 2*. For the analysis of the Cancer Dependency Map (DepMap) Common Essential genes, the 20Q1 Common Essential genes were downloaded from https://depmap.org/portal/download/. For receiver operating characteristic curve analysis, 'positives' were defined as genes with a K562 CRISPRi growth screen p-value <0.001 and $\gamma$<−0.05 (*Horlbeck et al., 2016a*), and 'negatives' were defined as non-targeting control guide pairs.

## Empirical sgRNA selection, incorporation of IBC, and validation of finalized dual-sgRNA CRISPRi libraries

While the pilot dual-sgRNA library dJR020 enabled validation of the dual-sgRNA strategy, finalized dual-sgRNA libraries were designed with additional considerations. An expanded set of aggregated CRISPR screen data was used to optimize guide selection, including data from screens previously published in *Adamson et al., 2016*; *Brown et al., 2022*; *Das et al., 2021*; *Hein and Weissman, 2022*; *Hickey et al., 2020*; *Horlbeck et al., 2016b*; *Jost et al., 2020*; *Jost et al., 2017*; *Lou et al., 2019*; *Martinko et al., 2018*; *Ramkumar et al., 2020*; *Shao et al., 2022*; *Shao et al., 2018*; *Tian et al., 2021*; *Tian et al., 2019*; *Torres et al., 2019*; *Le vasseur et al., 2021*; *Supplementary file 3*. Optimal sgRNAs targeting each gene were selected using an updated set of rules. First, sgRNAs containing a BsmBI target sequence (CGTCTC or GAGACG) were removed to avoid dropout during cloning. Second, each transcript per gene in *Horlbeck et al., 2016a*, was targeted independently. Genes were separated into three tiers, similar to the tiers described for the pilot library but with additional considerations:

Tier 1 (n=662 genes): For genes essential for growth in the K562 CRISPRi screen data (p-value <0.001 and $\gamma < −0.2$) (*Horlbeck et al., 2016a*), sgRNAs were ranked by their growth phenotypes (calculated relative to the best-performing sgRNA targeting each gene per screen in which the gene was a significant hit at FDR 0.05).

Tier 2 (n=4033 genes): The ranking strategy used to generate the pilot library (dJR020) included any gene identified as a significant hit in any previous CRISPRi screen for empirical guide selection and

as such did not control for the increased chance that a gene may score as a false positive in a screen as the number of screens increases (the equivalent of multiple comparisons). To control for such false positives, the 320 olfactory genes served as a negative control set. None of the 320 olfactory genes were a significant hit (FDR 0.05 based on MAGeCK RRA p-value; *Li et al., 2014*) in greater than four previous CRISPRi screens. Therefore, as a first cutoff, any gene that was identified as a significant hit in five or more previous CRISPRi screens, regardless of the strength of the phenotype, was included in this tier.

This cutoff misses genes that score strongly, and as such are high-confidence hits, in a small number of screens. To also include such genes, each gene that was a significant hit (FDR 0.05 based on MAGeCK RRA p-value; *Li et al., 2014*) in one to four screens was assigned a score based on the maximum absolute value discriminant score (calculated as the $-\log_{10}$ p-value multiplied by the mean $z$-scored phenotype of the top three sgRNAs), summed across screens in which the gene scored as a hit. As a comparison, this same score was calculated for olfactory genes. Genes were included in this tier if the discriminant score was greater than a threshold calculated from the olfactory gene scores for the same number of screens in which a gene was identified as a hit.

For all genes included in this tier, sgRNAs were ranked by the average of phenotypes across screens in which the gene was identified as a hit. Only sgRNAs that were identified as a hit at FDR <0.01 in at least one screen were ranked. sgRNA phenotypes were calculated relative to the best performing sgRNA targeting each gene per screen in which the gene was a significant hit at FDR 0.05.

Tier 3 (n=14,493 genes): For all other genes, sgRNAs were ranked by the regression scores in hCRISPRi v2.1 (*Horlbeck et al., 2016a*).

Using this ranking scheme, we selected the first and second ranked sgRNAs for a dual-sgRNA/single-element-per-gene sublibrary (hCRISPRi_dual_1_2), the third and fourth ranked sgRNAs for a second dual-sgRNA/single-element-per-gene sublibrary (hCRISPRi_dual_3_4), and the fifth and sixth ranked sgRNAs for a final dual-sgRNA/single-element-per-gene sublibrary (hCRISPRi_dual_5_6). Each library also contains a set of non-targeting control dual sgRNAs representing 5% of the total library elements. A list of sgRNA targeting sequences for all libraries is available in *Supplementary file 4*.

IBCs were incorporated between the tandem sgRNA cassettes in the dual-sgRNA library in four steps. First, a library of 215 8-nucleotide IBCs were designed with a Hamming distance ≥4 and between 25% and 75% GC content (*Supplementary file 5*). Second, the library of IBCs were cloned into pJR98 in an arrayed format. pJR98 was digested by AscI and ssDNA oligo donors of the sequence 5' CTCTTCCTGCCCGACCTTGGGG – reverse complement IBC – CAGCGCCATAGC TGAGTGTAGATTCGAGC – 3' were cloned into the vector using NEBuilder HiFI DNA Assembly Master Mix (NEB). Third, the library of cloned IBCs were Sanger verified and pooled at a equimolar ratio for all barcodes. Fourth, the library was cloned into the dual-sgRNA library by BsmBI digestion and ligation. Sequencing was performed on a NovaSeq 6000 (Illumina) using a 19 bp Read 1, 19 bp Read 2, 8 bp Index Read 1, and 8 bp Index Read 2 with custom sequencing primers as described in the sequencing library preparation protocol available at https://www.jostlab.org/resources/ and https://weissman.wi.mit.edu/resources/. Demultiplexing on only the i5 index using the i7 index (IBC) as a read was performed as detailed: https://gist.github.com/sumeetg23/a064 a36801d2763e94da2e191699fb9f. The human CRISPRi dual-sgRNA libraries with IBCs are available from Addgene (hCRISPRi_dual_1_2, Addgene #187246; hCRISPRi_dual_3_4, Addgene #187247; hCRISPRi_dual_5_6, Addgene #187248).

## Perturb-seq comparison of dual-sgRNA libraries versus Dolcetto

Direct capture Perturb-seq (*Replogle et al., 2020*) was used to directly compare the knockdown produced by the dual-sgRNA libraries versus the Dolcetto Set A CRISPRi library. N=128 genes were randomly selected from the 4000 most highly expressed genes in K562 cells based on RNA-seq (https://www.encodeproject.org/experiments/ENCSR000AEL/). Two parallel libraries were cloned: a library containing the three dual-sgRNA elements targeting each gene and a library containing the three Dolcetto Set A guides targeting each gene, plus non-targeting control guides. For Dolcetto sgRNAs, the 5' base was replaced with a G to enable expression from the U6 promoter. The Dolcetto single-sgRNA library was cloned as described above into pJR101 guide expression vector containing a Perturb-seq capture sequence in stem loop 2. The dual-sgRNA library cloned as described above into pJR101 with a pJR98 insert cassette containing a Perturb-seq capture sequence in stem loop 2

of guide B. After library verification by sequencing, lentivirus was prepared in HEK293T as described above.

For Perturb-seq, CRISPRi K562 cells expressing dCas9-Kox1 (*Gilbert et al., 2014*) were spinfected (1000 × *g*) with polybrene (8 μg/mL) with lentivirus from both libraries in parallel. Throughout the screen, cells were maintained at a density between 250,000 and 1,000,000 cells/mL and 1000× coverage per library element. On day 3 post-transduction, an infection rate of 5% was measured for both screens, and infected cells were sorted by FACS (BD FACS Aria). On day 7 post-transduction, cells were prepared for single-cell RNA-sequencing as detailed in the 10x Genomics Single Cell Protocols Cell Preparation Guide (10x Genomics, CG00053 Rev C) and separated into droplet emulsions using the Chromium Controller (10x Genomics) with Chromium Single-Cell 3′ Gel Beads v3.1 (10x Genomics, PN-1000121 and PN-1000120) across 12 lanes/gemgroups with the goal of recovering ~15,000 cells per GEM group before filtering. Sequencing libraries were prepared following the 10x Genomics Chromium Single Cell 3′ Reagent Kits User Guide (v3.1 Chemistry) with Feature Barcoding technology for CRISPR Screening (CG000205; Rev C). Libraries were sequenced on a NovaSeq 6000 (Illumina) according to the 10x Genomics User Guide.

After sequencing, mRNA and sgRNA counts were obtained from Cell Ranger 4.0.0 software (10x Genomics). To assign guides to cells, we used a Poisson-Gaussian mixture model as previously described (*Replogle et al., 2020*). Only cells bearing a single Dolcetto sgRNA or a single dual-sgRNA guide B sgRNA were used for downstream calculation of CRISPRi efficacy. For each guide, the on-target knockdown was calculated as the fraction of mRNA remaining (target gene expression in targeting cells relative to cells bearing non-targeting control guides).

## Design of genome-wide human CRISPRa, mouse CRISPRi, and mouse CRISPRa libraries

To design our genome-wide human CRISPRa (*Supplementary file 9*), mouse CRISPRi (*Supplementary file 10*), and mouse CRISPRa libraries (*Supplementary file 11*), sgRNAs targeting each gene were selected from our previously published Horlbeck et al. predicted rankings (*Horlbeck et al., 2016a*) due to the paucity of empirical data. sgRNAs were then filtered to exclude sgRNAs containing BsmBI cut sites and ranked by predicted activities. Libraries were created by combining the first and second ranked sgRNAs, the third and fourth ranked sgRNAs, and the fifth and sixth ranked sgRNAs. Each library also contains a set of non-targeting control dual sgRNAs representing 5% of the total library elements. The human CRISPRa libraries were synthesized as an oligonucleotide pool (Twist Biosciences) cloned according to the protocol available at https://www.jostlab.org/resources/ and https://weissman.wi.mit.edu/resources/ using pJR104 as a base vector and IBCs as described above. The human CRISPRa dual-sgRNA libraries with IBCs are available from Addgene (hCRISPRa_dual_1_2, Addgene #187249; hCRISPRa_dual_3_4, Addgene #187250; hCRISPRa_dual_5_6, Addgene #187251). The mouse libraries are only provided as in silico designs and have not been synthesized or cloned.

## Design and cloning of constructs for CRISPRi effector expression

All CRISPRi effectors were cloned into a lentiviral backbone containing a UCOE and an SFFV promoter (pMH0001, Addgene #85969). Briefly, dCas9, effector domains, linker domains, and GFP were PCR-amplified and inserted into backbone linearized by digest with MluI and NotI using Gibson assembly (NEBuilder HiFI DNA Assembly Master Mix, NEB). P2A sequences were incorporated into primer overhangs. The following additional considerations were incorporated into the final construct designs:

1. For KRAB from Kox1, the KRAB(KOX1) domain from dCas9-BFP-KRAB (Addgene #46911) was fused to the C-terminus of dCas9, because C-terminal fusions of KRAB(KOX1) have historically produced the highest activity, linked by an 80-amino acid linker (XTEN80). XTEN80-KRAB(KOX1) was synthesized as a gBlock (IDT). We chose XTEN80 because we previously found that inclusion of a linker increases activity and the original dCas9-BFP-KRAB(KOX1) construct (*Gilbert et al., 2013*) underwent proteolytic cleavage between dCas9 and KRAB(KOX1) in some cell types, giving rise to free dCas9, a dominant negative for CRISPRi. The final construct is dCas9-XTEN80-KRAB(KOX1) or dCas9-Kox1 for short.
2. KRAB(ZIM3) was fused to the N-terminus of dCas9 with a 6-amino acid GS linker, which had produced the highest activity in a previous report, including when compared to C-terminal fusions (*Alerasool et al., 2020*). KRAB(ZIM3) was PCR-amplified from pLX303-ZIM3-KRAB-dCas9

(Addgene #154472). The final construct is KRAB(ZIM3)-dCas9 or Zim3-dCas9 for short. Note that this construct contains an additional NLS between Zim3 and dCas9.

3. For SID4x, SID4x was fused to the N-terminus of dCas9-XTEN80-KRAB(Kox1), because SID4x had previously only been evaluated for CRISPRi in the context of a dual fusion (*Carleton et al., 2017*). A shorter 16-aa linker (XTEN16) was included between SID4x and dCas9, which has been a sufficient linker length at the N-terminus in the past. SID4x was amplified from a construct generously donated by the Aifantis lab (New York University). The final construct is SID4x-XTEN16-dCas9-XTEN80-KRAB(KOX1) or SID-dCas9-Kox1 for short.

4. For MeCP2, the previously reported dCas9-KRAB(Kox1)-MeCP2 construct (Addgene #110821; *Yeo et al., 2018*) was PCR-amplified and transferred into the common backbone, giving rise to dCas9-Kox1-MeCP2. Note that this construct contains no linker between dCas9 and KRAB(Kox1), such that the KRAB(Kox1) domain may be largely inactive, and that the dCas9 uses different codons. We separately also generated a construct in which we fused MeCP2 to the C-terminus of the dCas9-XTEN80-KRAB(KOX1) construct. We observed similar growth defects and non-specific effects on the transcriptome using this construct.

Additional constructs with expression driven by an EF1α promoter were generated by performing analogous assemblies in the pMH0006 backbone (Addgene #135448). Constructs with expression driven by CMV or EFS promoters were generated by replacing the SFFV promoter in existing constructs. Constructs in which effector expression is marked with BFP, mCherry, or hygromycin resistance were generated by assembling with PCR products containing the desired markers. Constructs in which expression of the fluorescent protein is linked by an IRES from encephalomyocarditis virus were generated by incorporating a PCR fragment generated from pHR-TRE3G-TUBB-IRES-mCherry (*Jost et al., 2017*) instead of the P2A site. Constructs in which EGFP is flanked by loxP sites were generated by PCR-amplifying EGFP with primers containing loxP 2272 sequences (ATAACTTCGTATAAaGTATc CTATACGAAGTTAT). The amplicon was inserted by Gibson Assembly into pJB069 or pJB109 linearized by digestion with NotI and AsiSI. Finally, constructs in which the fluorescent proteins are constitutively linked to dCas9 were generated by omitting the P2A sequence from primer overhangs. A full list of generated constructs is included in *Supplementary file 7*. All constructs are available on Addgene.

## Evaluation of effects of CRISPRi effectors on growth and transcription

K562 cell lines stably expressing CRISPRi effectors from an SFFV promoter linked to GFP via P2A were generated by lentiviral transduction and FACS. Each effector expression construct was transduced in triplicate in parallel with all other constructs; 100,000 GFP-positive cells per replicate were isolated by FACS on a Sony SH800S Cell Sorter 5 days after transduction and allowed to recover.

To generate RNA-seq libraries of cells expressing each effector, $1 \times 10^6$ cells were harvested for each sample 6 days after FACS by centrifugation at $300 \times g$ for 5 min and flash-frozen in a dry ice and ethanol bath. RNA was extracted using the Direct-zol RNA Miniprep kit (Zymo Research) and quantified using the Qubit RNA BR Assay Kit (Life Technologies). RNA-seq libraries were prepared by the Whitehead Genome Technology Core facility using the Roche Diagnostics KAPA mRNA HyperPrep Kit. Paired-end 100 sequencing was performed on a NovaSeq (Illumina).

To evaluate growth of CRISPRi effector-expressing K562 cells, a reference population of K562 cells stably expressing mCherry was generated by lentiviral transduction of pU6-sgRNA EF1Alpha-puro-T2A-mCherry (a gift from Gregory Ow and Eric Collisson, UCSF) and FACS. This was conducted in parallel with the generation of CRISPRi effector-expressing cells. Seven days after sorting, ~125,000 cells per GFP-sorted population (different CRISPRi effectors) were mixed with ~125,000 mCherry-sorted cells (reference population). The ratio of mCherry-positive to mCherry-negative cells was read out immediately after mixing and periodically for the next 19 days by flow cytometry on an Attune NxT (Thermo Fisher).

Growth of HepG2, HuTu-80, and HT29 cells expressing Zim3-dCas9 was performed analogously, except that Zim3-dCas9 was marked with mCherry and the reference population was marked with BFP, by transduction with pCRISPRi/a-v2 containing a Gal4-4 sgRNA. Transduced cells were sorted on a FACSMelody cell sorter (BD Biosciences). Seven days after sorting, ~150,000 cells per mCherry-sorted population (Zim3-dCas9) were mixed with ~150,000 BFP-sorted cells (reference population) and seeded over three independent wells. The ratio of BFP-positive to BFP-negative cells was read out periodically for the next 10–14 days by flow cytometry on a FACSymphony A1 (BD Biosciences).

## RNA-seq data analysis

Sequencing reads were aligned strand-specifically to the human genome (GRCh38) and then aggregated by gene using only reads uniquely mapped to the reverse strand using the spliced read aligner STAR (*Dobin et al., 2013*), version 2.7.9, against an index containing features from Ensembl release 98/GENCODE v32 (downloaded from 10x Genomics reference 2020-A). Replicate sample 2 for cells expressing dCas9-Kox1 had substantially fewer reads than expected and was excluded from analysis. For clustering analysis, transcript counts were normalized to transcripts per million for each sample, filtered for the 2000 most highly expressed genes on average, and clustered using the Ward variance minimization algorithm implemented in scipy version 1.6.2. Differential expression analysis was carried out on gene counts using DESeq2 (*Love et al., 2014*). For *Figure 2E*, transcript counts were not filtered. The trends for numbers of differentially expressed genes were equivalent when only including genes with an average count >2 across all samples.

## Selection and cloning of individual sgRNAs

Strong sgRNAs against essential genes or cell surface markers were selected from the hCRISPRi-v2 library (*Horlbeck et al., 2016a*; *Nuñez et al., 2021*). Intermediate-activity sgRNAs were selected either from the hCRISPRi-v2 library or by incorporating defined mismatches in strong sgRNAs (*Jost et al., 2020*). All sgRNA sequences used for individual evaluation are listed in *Supplementary file 8*.

Individual sgRNA expression constructs were cloned as described previously (*Gilbert et al., 2014*). Briefly, two complementary oligonucleotides (IDT), containing the sgRNA targeting region as well as overhangs matching those left by restriction digest of the vector with BstXI and BlpI, were annealed and ligated into pCRISPRia-v2 (pU6-sgRNA EF1Alpha-puro-T2A-BFP with two SbfI sites flanking the sgRNA expression cassette, Addgene #84832; *Horlbeck et al., 2016a*) or pU6-sgRNA EF1Alpha-puro-T2A-mCherry (a gift from Gregory Ow and Eric Collisson, UCSF; *Jost et al., 2020*) digested with BstXI (NEB or Thermo Fisher Scientific) and BlpI (NEB) or Bpu1102I (Thermo Fisher Scientific). The ligation product was transformed into Stellar chemically competent *Escherichia coli* cells (Takara Bio) and plasmid was prepared following standard protocols. The resulting sgRNA expression vectors were individually packaged into lentivirus as described above.

## Evaluation of individual sgRNA phenotypes

Effects of sgRNAs targeting essential genes on cell growth were measured in internally controlled growth assays by transducing cells with mCherry-marked sgRNA expression constructs at multiplicity of infection <0.5 (15–40% infected cells) and measuring the fraction of sgRNA-expressing cells 3–12 days after transduction as mCherry-positive cells by flow cytometry on an Attune NxT (Thermo Fisher). All experiments were performed in duplicates from the infection step.

Effects of sgRNAs on expression levels of cell surface proteins were measured by flow cytometry. K562 or Jurkat cell lines expressing CRISPRi effectors of interest were infected with lentivirus containing sgRNA expression vectors by centrifugation at 1000 × *g* for 1 hr in 24-well plates in the presence of 8 µg/mL polybrene. RPE1, HepG2, HuTu-80, and HT29 cell lines expressing Zim3-dCas9 were infected with lentivirus containing sgRNA expression vectors for 24 hr in the presence of 8 µg/mL polybrene. Six to 14 days after transduction, cells were harvested by centrifugation (suspension cells) or trypsin-free detachment (adherent cells; mechanical detachment or EDTA), washed once in flow cytometry buffer (PBS with 5% (v/v) FBS), and stained at room temperature for 15–30 min with APC-conjugated antibodies targeting CD55 (clone JS11, BioLegend 311311, RRID:AB_2075857), CD81 (clone 5A6, BioLegend 349509, RRID:AB_2564020), CD151 (clone 50-6, BioLegend 350405, RRID:AB_10661726), CD29 (clone TS2/16, BioLegend 303007, RRID:AB_314323), or B2M (clone 2M2, BioLegend 316312, RRID:AB_10641281) diluted 1:100 in flow cytometry buffer. After staining, cells were washed twice in 200 µL flow cytometry buffer and resuspended in flow cytometry buffer for measurement on an Attune NxT (Thermo Fisher), LSR-II (BD Biosciences), or FACSymphony A3 (BD Biosciences).

Optimal dilutions for each antibody were determined by testing 1:20, 1:100, and 1:500 antibody titrations on K562 cells with epitope-targeting or non-targeting sgRNAs and choosing the titration with the maximum signal difference.

Flow cytometry data were analyzed using FlowCytometryTools 0.5.0 (https://eyurtsev.github.io/FlowCytometryTools/) and python 3.8. Briefly, the data were gated for cells (FSC-A versus SSC-A), FSC

singlets (FSC-W versus FSC-H for data recorded on an Attune NxT and FSC-W versus FSC-A for data recorded on an LSR-II), SSC singlets (SSC-W versus SSC-H for data recorded on an Attune NxT and SSC-W versus SSC-A for data recorded on an LSR-II), and sgRNA-expressing cells (BFP- or mCherry-positive, depending on the experiment). Background APC fluorescence intensity from unstained cells or cells stained with an APC-conjugated Mouse IgG1, κ isotype control (BioLegend clone MOPC-21) was subtracted to correct for background fluorescence. Knockdown was quantified using median background-corrected APC fluorescence intensity in cells expressing a targeting sgRNA relative to intensity in cells expressing a non-targeting control sgRNA, with the exception of the Jurkat and RPE1 experiments, for which knockdown was quantified using median background-corrected APC fluorescence intensity in cells expressing a targeting sgRNA relative to intensity in cells not expressing an sgRNA in the same well.

## Acknowledgements

We thank B Adamson, J Nuñez, J Hussmann, M Horlbeck, E Chow, S Gupta, and F Urnov for helpful discussions. We thank M de Vera, S Sinha, C Muresan, and J Kanter for lab and administrative support. We thank I Aifantis (NYU) for donating a vector containing SID4x and G Ow and E Collisson (UCSF) for donating the mCherry-marked sgRNA expression vector. Sequencing was in part performed at the UCSF Center for Advanced Technology, supported by UCSF PBBR, RRP IMIA, and NIH 1S10OD028511-01 grants. We thank the Whitehead Institute Flow Cytometry Core and the Harvard Immunology Flow Cytometry Core Facility for access to FACS machines and flow cytometers and the Whitehead Institute Genome Technology Core for performing bulk RNA-seq library preparation. Funding: This work was supported in part by the National Institutes of Health (grants R00GM130964 to MJ and RM1HG009490-01 to JSW), Springer Nature Global Grant for Gut Health 1772808 (MJ), a Charles H Hood Foundation Child Health Research Award (MJ), DARPA HR0011-19-2-0007 (JSW), the Ludwig Center at MIT (JSW), and the Chan Zuckerberg Initiative (JSW). JMR was supported by NIH F31 Ruth L Kirschstein National Research Service Award NS115380. KL was supported by NIH F30 Ruth L Kirschstein National Research Service Award AG066418. BJR was supported by Harvard Bacteriology PhD Training Program T32 AI132120. AG was supported by Human Frontier Science Program grant 2019L/LT000858. LAG is funded by an NIH New Innovator Award (DP2 CA239597), a Pew-Stewart Scholars for Cancer Research award, and the Goldberg-Benioff Endowed Professorship in Prostate Cancer Translational Biology. MK was supported by the Chan Zuckerberg Initiative Ben Barres Early Career Acceleration Award. JSW is a Howard Hughes Medical Institute Investigator.

## Additional information

### Competing interests

Joseph M Replogle: consults for Maze Therapeutics and Waypoint Bio. Thomas M Norman: consults for Maze Therapeutics. The Regents of the University of California with TMN, MJ, LAG, and JSW as inventors have filed patent applications related to CRISPRi/a screening and Perturb-seq. Luke A Gilbert: declares outside interest in Chroma Medicine. The Regents of the University of California with TMN, MJ, LAG, and JSW as inventors have filed patent applications related to CRISPRi/a screening and Perturb-seq. LAG, MK, and JSW are inventors on US Patent 11,254,933 related to CRISPRi/a screening. Martin Kampmann: serves on the Scientific Advisory Boards of Engine Biosciences, Casma Therapeutics, Cajal Neuroscience, and Alector, and is an advisor to Modulo Bio and Recursion Therapeutics. LAG, MK, and JSW are inventors on US Patent 11,254,933 related to CRISPRi/a screening. Jonathan S Weissman: declares outside interest in 5 AM Venture, Amgen, Chroma Medicine, KSQ Therapeutics, Maze Therapeutics, Tenaya Therapeutics, Tessera Therapeutics, and Third Rock Ventures. The Regents of the University of California with TMN, MJ, LAG, and JSW as inventors have filed patent applications related to CRISPRi/a screening and Perturb-seq. LAG, MK, and JSW are inventors on US Patent 11,254,933 related to CRISPRi/a screening. Marco Jost: consults for Maze Therapeutics and Gate Bioscience. The Regents of the University of California with TMN, MJ, LAG, and JSW as inventors have filed patent applications related to CRISPRi/a screening and Perturb-seq. The other authors declare that no competing interests exist.

## Funding

| Funder | Grant reference number | Author |
|---|---|---|
| National Institutes of Health | R00GM130964 | Marco Jost |
| National Institutes of Health | RM1HG009490-01 | Jonathan S Weissman |
| Springer Nature Global Grant for Gut Health | 1772808 | Marco Jost |
| Charles H. Hood Foundation | Child Health Research Award | Marco Jost |
| Defense Advanced Research Projects Agency | HR0011-19-2-0007 | Jonathan S Weissman |
| Ludwig Center for Molecular Oncology | | Jonathan S Weissman |
| Chan Zuckerberg Initiative | | Jonathan S Weissman |
| National Institutes of Health | F31NS115380 | Joseph M Replogle |
| National Institutes of Health | F30AG066418 | Kun Leng |
| National Institutes of Health | T32AI132120 | Baylee J Russell |
| Human Frontier Science Program | 2019L/LT000858 | Alina Guna |
| Chan Zuckerberg Initiative | Ben Barres Early Career Acceleration Award | Martin Kampmann |
| Howard Hughes Medical Institute | Investigator | Jonathan S Weissman |
| National Institutes of Health | DP2 CA239597 | Luke A Gilbert |
| Pew Charitable Trusts | Pew-Stewart Scholars for Cancer Research Award | Luke A Gilbert |
| UCSF School of Medicine | Goldberg-Benioff Endowed Professorship in Prostate Cancer Translational Biology | Luke A Gilbert |

The funders had no role in study design, data collection and interpretation, or the decision to submit the work for publication.

## Author contributions

Joseph M Replogle, Conceptualization, Data curation, Formal analysis, Investigation, Methodology, Resources, Software, Visualization, Writing – original draft, Writing – review and editing, Generated and tested dual-sgRNA libraries, Created dual-sgRNA cloning and sequencing protocols, Analyzed screen and Perturb-seq data, Co-wrote the manuscript; Jessica L Bonnar, Investigation, Methodology, Validation, Visualization, Writing – original draft, Writing – review and editing, Generated Zim3-dCas9 expression constructs, Generated K562 cell lines, Assayed effects on growth and transcription, Measured knockdown in K562 cells, Co-wrote the manuscript; Angela N Pogson, Investigation, Methodology, Validation, Visualization, Writing – review and editing, Cloned dual-sgRNA libraries, Performed screens and Perturb-seq, Generated sequencing libraries, Generated and validated Jurkat and RPE1 cells expressing Zim3-dCas9, Provided input on the manuscript; Christina R Liem, Investigation, Resources, Writing – review and editing, Generated effector expression constructs, Performed preliminary tests, Provided input on the manuscript; Nolan K Maier, Investigation, Resources, Writing – review and editing, Generated additional Zim3-dCas9 expression constructs, Generated and assayed additional cells expressing Zim3-dCas9, Provided input on the manuscript; Yufang Ding, Investigation, Writing – review and editing, Generated and assayed additional cells expressing Zim3-dCas9,

Provided input on the manuscript; Baylee J Russell, Investigation, Writing – review and editing, Generated and assayed additional cells expressing Zim3-dCas9, Provided input on the manuscript; Xingren Wang, Investigation, Writing – review and editing, Generated and assayed additional cells expressing Zim3-dCas9, Provided input on the manuscript; Kun Leng, Methodology, Resources, Writing – review and editing, Provided screen data, Helped with data aggregation, Provided input on the manuscript; Alina Guna, Investigation, Resources, Writing – review and editing, Generated Zim3-dCas9(Hygro), Validated the corresponding K562 cell line, Provided input on the manuscript; Thomas M Norman, Investigation, Writing – review and editing, Helped with effector testing, Provided input on the manuscript; Ryan A Pak, Investigation, Writing – review and editing, Generated IRES-linked effector expression constructs, Provided input on the manuscript; Daniel M Ramos, Resources, Writing – review and editing, Provided unpublished screen data, Provided input on the manuscript; Michael E Ward, Resources, Supervision, Writing – review and editing, Provided unpublished screen data, Provided input on the manuscript; Luke A Gilbert, Resources, Writing – review and editing, Provided unpublished screen data, Edited the manuscript; Martin Kampmann, Funding acquisition, Resources, Supervision, Writing – review and editing, Provided screen data, Helped with data aggregation, Edited the manuscript; Jonathan S Weissman, Conceptualization, Supervision, Funding acquisition, Methodology, Project administration, Writing – review and editing, Supervised all work, Obtained funding, Edited the manuscript; Marco Jost, Conceptualization, Data curation, Formal analysis, Funding acquisition, Methodology, Project administration, Supervision, Visualization, Writing – original draft, Writing – review and editing, Supervised all work, Obtained funding, Co-wrote the manuscript

**Author ORCIDs**
Joseph M Replogle  http://orcid.org/0000-0003-1832-919X
Jessica L Bonnar  http://orcid.org/0000-0001-5531-4849
Angela N Pogson  http://orcid.org/0000-0002-6927-2456
Christina R Liem  http://orcid.org/0000-0003-2744-6312
Nolan K Maier  http://orcid.org/0000-0001-6103-6726
Yufang Ding  http://orcid.org/0000-0001-6633-8192
Baylee J Russell  http://orcid.org/0000-0001-5284-3809
Michael E Ward  http://orcid.org/0000-0002-5296-8051
Luke A Gilbert  http://orcid.org/0000-0001-5854-0825
Martin Kampmann  http://orcid.org/0000-0002-3819-7019
Jonathan S Weissman  http://orcid.org/0000-0003-2445-670X
Marco Jost  http://orcid.org/0000-0002-1369-4908

**Decision letter and Author response**
Decision letter https://doi.org/10.7554/eLife.81856.sa1
Author response https://doi.org/10.7554/eLife.81856.sa2

## Additional files

### Supplementary files
• Supplementary file 1. Table with dual- and single-single guide RNA (sgRNA) sequences used for preliminary comparison.

• Supplementary file 2. Table with read counts and growth phenotypes from pilot screen.

• Supplementary file 3. Table with aggregated CRISPR interference (CRISPRi) single guide RNA (sgRNA) performance across screens.

• Supplementary file 4. Table with finalized dual-single guide RNA (sgRNA) human CRISPR interference (CRISPRi) libraries.

• Supplementary file 5. List of integration barcodes.

• Supplementary file 6. Table with Dolcetto versus dual-single guide RNA (sgRNA) Perturb-seq comparison.

• Supplementary file 7. Table of plasmids.

• Supplementary file 8. Table with sequences of single guide RNAs (sgRNAs) used for individual validation.

• Supplementary file 9. Table with dual-single guide RNA (sgRNA) human CRISPR activation

(CRISPRa) libraries.

• Supplementary file 10. Table with in silico dual-single guide RNA (sgRNA) mouse CRISPR interference (CRISPRi) libraries.

• Supplementary file 11. Table with in silico dual-single guide RNA (sgRNA) mouse CRISPR activation (CRISPRa) libraries.

• Supplementary file 12. Table with dual-single guide RNA (sgRNA) human CRISPR interference (CRISPRi) library targeting DepMap Common Essential genes.

• MDAR checklist

## Data availability

Sequencing data are available on NCBI GEO under accession number GSE205310 (Perturb-seq) and GSE205147 (bulk RNA-seq). sgRNA counts from CRISPRi screens are included as supplementary files. All data generated or analyzed during this study are included in the manuscript and supporting files.

The following datasets were generated:

| Author(s) | Year | Dataset title | Dataset URL | Database and Identifier |
|---|---|---|---|---|
| Replogle J, Pogson A, Weissman J, Jost M | 2022 | Comparison of dual sgRNA library versus Dolcetto by Perturb-seq | https://www.ncbi.nlm.nih.gov/geo/query/acc.cgi?acc=GSE205310 | NCBI Gene Expression Omnibus, GSE205310 |
| Replogle J, Bonnar J, Weissman J, Jost M | 2022 | Comparison of non-specific transcriptional effects of CRISPRi effector proteins by RNA-seq | https://www.ncbi.nlm.nih.gov/geo/query/acc.cgi?acc=GSE205147 | NCBI Gene Expression Omnibus, GSE205147 |

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

## Appendix 1

**Appendix 1—key resources table**

| Reagent type (species) or resource | Designation | Source or reference | Identifiers | Additional information |
|---|---|---|---|---|
| Cell line (Human) | K562 CRISPRi (BFP) | This paper | N/A | Stable cell line with SFFV Zim3-dCas9 P2A BFP |
| Cell line (Human) | K562 CRISPRi (GFP) | This paper | N/A | Stable cell line with SFFV Zim3-dCas9 P2A GFP |
| Cell line (Human) | K562 CRISPRi (no fluorescent marker) | This paper | N/A | Stable cell line with SFFV Zim3-dCas9 P2A Hygro |
| Cell line (Human) | RPE1 CRISPRi (Zim3) | *Replogle et al., 2022* | N/A | Stable cell line with SFFV Zim3-dCas9 P2A BFP |
| Cell line (Human) | Jurkat CRISPRi (Zim3) | This paper | N/A | Stable cell line with EF1alpha Zim3-dCas9 P2A mCherry |
| Cell line (Human) | HepG2 CRISPRi | This paper | N/A | Stable cell line with EF1alpha Zim3-dCas9 P2A mCherry |
| Cell line (Human) | HT29 CRISPRi | This paper | N/A | Stable cell line with EF1alpha Zim3-dCas9 P2A mCherry |
| Cell line (Human) | HuTu-80 CRISPRi | This paper | N/A | Stable cell line with EF1alpha Zim3-dCas9 P2A mCherry |
| Antibody | Anti-human CD55 (clone JS11, mouse monoclonal), APC | BioLegend | Cat#: 311311; RRID:AB_2075857 | Flow cytometry; 1:100 in PBS with 5% (v/v) FBS |
| Antibody | Anti-human CD81 (clone 5A6, mouse monoclonal), APC | BioLegend | Cat#: 349509; RRID:AB_2564020 | Flow cytometry; 1:100 in PBS with 5% (v/v) FBS |
| Antibody | Anti-human CD151 (clone 50–6, mouse monoclonal), APC | BioLegend | Cat#: 350405; RRID:AB_10661726 | Flow cytometry; 1:100 in PBS with 5% (v/v) FBS |
| Antibody | Anti-human CD29 (clone TS2/16, mouse monoclonal), APC | BioLegend | Cat#: 303007; RRID:AB_314323 | Flow cytometry; 1:100 in PBS with 5% (v/v) FBS |
| Antibody | Anti-human B2M (clone 2M2, mouse monoclonal), APC | BioLegend | Cat#: 316312; RRID:AB_10641281 | Flow cytometry; 1:100 in PBS with 5% (v/v) FBS |
| Antibody | Mouse IgG1, κ Isotype Ctrl (clone MOPC-21, mouse monoclonal), APC | BioLegend | Cat#: 981806 | Flow cytometry; 1:100 in PBS with 5% (v/v) FBS |
| Recombinant DNA reagent | Plasmid pJB120_pHR-UCOE-SFFV-EGFP | This paper | Addgene: 188900 | Further information in *Supplementary file 7* |
| Recombinant DNA reagent | Plasmid pJB074_pHR-UCOE-SFFV-dCas9-HA-2xNLS-P2A-EGFP | This paper | Addgene: 188898 | Further information in *Supplementary file 7* |
| Recombinant DNA reagent | Plasmid pJB069_pHR-UCOE-SFFV-Zim3-NLS-dCas9-HA-2xNLS-P2A-EGFP | This paper | Addgene: 188899 | Further information in *Supplementary file 7* |
| Recombinant DNA reagent | Plasmid pCL63_pHR-UCOE-SFFV-SID4x-dCas9-HA-2xNLS-XTEN80-KRAB(Kox1)-P2A-EGFP | This paper | Addgene: 188901 | Further information in *Supplementary file 7* |

*Appendix 1 Continued on next page*

*Appendix 1 Continued*

| Reagent type (species) or resource | Designation | Source or reference | Identifiers | Additional information |
|---|---|---|---|---|
| Recombinant DNA reagent | Plasmid pCL51_pHR-UCOE-SFFV-dCas9-NLS-KRAB(Kox1)-MeCP2-P2A-EGFP | This paper | Addgene: 188902 | Further information in *Supplementary file 7* |
| Recombinant DNA reagent | Plasmid pCL35_pHR-UCOE-SFFV-dCas9-HA-2xNLS-XTEN80-KRAB(Kox1)-P2A-EGFP | This paper | Addgene: 188765 | Further information in *Supplementary file 7* |
| Recombinant DNA reagent | Plasmid pAG389_pHR-UCOE-SFFV-Zim3-NLS-dCas9-HA-2xNLS-P2A-Hygro | This paper | Addgene: 188768 | Further information in *Supplementary file 7* |
| Recombinant DNA reagent | Plasmid pNM1130_pHR-UCOE-EF1a-Zim3-NLS-dCas9-HA-2xNLS-loxP-P2A-EGFP-loxP | This paper | Addgene: 188773 | Further information in *Supplementary file 7* |
| Recombinant DNA reagent | Plasmid pNM1129_pHR-UCOE-SFFV-Zim3-NLS-dCas9-HA-2xNLS-loxP-P2A-EGFP-loxP | This paper | Addgene: 188774 | Further information in *Supplementary file 7* |
| Recombinant DNA reagent | Plasmid pNM1128_pHR-UCOE-EF1a-Zim3-NLS-dCas9-HA-2xNLS-mTagBFP2 | This paper | Addgene:188775 | Further information in *Supplementary file 7* |
| Recombinant DNA reagent | Plasmid pNM1127_pHR-UCOE-SFFV-Zim3-NLS-dCas9-HA-2xNLS-mTagBFP2 | This paper | Addgene: 188776 | Further information in *Supplementary file 7* |
| Recombinant DNA reagent | Plasmid pNM1125_pHR-UCOE-EF1a-Zim3-NLS-dCas9-HA-2xNLS-P2A-mTagBFP2 | This paper | Addgene:188777 | Further information in *Supplementary file 7* |
| Recombinant DNA reagent | Plasmid pNM1124_pHR-UCOE-EF1a-Zim3-NLS-dCas9-HA-2xNLS-P2A-EGFP | This paper | Addgene: 188778 | Further information in *Supplementary file 7* |
| Recombinant DNA reagent | Plasmid pNM1123_pHR-UCOE-SFFV-Zim3-NLS-dCas9-HA-2xNLS-P2A-mCherry | This paper | Addgene: 188779 | Further information in *Supplementary file 7* |
| Recombinant DNA reagent | Plasmid pJB109_pHR-UCOE-EF1a-Zim3-NLS-dCas9-HA-2xNLS-P2A-mCherry | This paper | Addgene: 188766 | Further information in *Supplementary file 7* |
| Recombinant DNA reagent | Plasmid pJB108_pHR-UCOE-SFFV-Zim3-NLS-dCas9-HA-2xNLS-P2A-mTagBFP2 | This paper | Addgene: 188767 | Further information in *Supplementary file 7* |
| Recombinant DNA reagent | Plasmid pRAP0006_pHR-UCOE-EF1a-dCas9-HA-2xNLS-XTEN80-KRAB(Kox1)-IRES-mCherry | This paper | Addgene: 188769 | Further information in *Supplementary file 7* |

*Appendix 1 Continued on next page*

*Appendix 1 Continued*

| Reagent type (species) or resource | Designation | Source or reference | Identifiers | Additional information |
|---|---|---|---|---|
| Recombinant DNA reagent | Plasmid pRAP0003_pHR-UCOE-SFFV-dCas9-HA-2xNLS-XTEN80-KRAB(Kox1)-IRES-mCherry | This paper | Addgene: 188770 | Further information in *Supplementary file 7* |
| Recombinant DNA reagent | Plasmid pCL75_pHR-UCOE-EFS-dCas9-HA-2xNLS-XTEN80-KRAB(Kox1)-P2A-EGFP | This paper | Addgene: 188771 | Further information in *Supplementary file 7* |
| Recombinant DNA reagent | Plasmid pCL74_pHR-UCOE-CMV-dCas9-HA-2xNLS-XTEN80-KRAB(Kox1)-P2A-EGFP | This paper | Addgene: 188772 | Further information in *Supplementary file 7* |
| Recombinant DNA reagent | Plasmid pCRISPRia-v2 (parent vector) | DOI: 10.7554/eLife.19760 | Addgene: 84832 | |
| Recombinant DNA reagent | Plasmid pU6-sgRNA EF1alpha-puro-T2A-mCherry | This paper | Addgene: 188780 | Further information in *Supplementary file 7* |
| Recombinant DNA reagent | pJR98 | This paper | Addgene: 187239 | CR3 constant region – hU6 sgRNA promoter flanked by BsmBI sites. Further information in *Supplementary file 7* |
| Recombinant DNA reagent | pJR100 | This paper | Addgene: 187240 | Lentiviral sgRNA vector for Perturb-seq with mU6 sgRNA promoter, CR1 constant region with CS1 capture sequence in stem loop, and UCOE EF1alpha driving PURO-BFP marker expression. Further information in *Supplementary file 7* |
| Recombinant DNA reagent | pJR101 | DOI: 10.1016/j.cell.2022.05.013/ this paper | Addgene: 187241 | Lentiviral sgRNA vector for Perturb-seq with mU6 sgRNA promoter, CR1 constant region with CS1 capture sequence in stem loop, and UCOE EF1alpha driving PURO-GFP marker expression. Further information in *Supplementary file 7* |
| Recombinant DNA reagent | pJR103 | This paper | Addgene: 187242 | Lentiviral sgRNA vector with mU6 sgRNA promoter, CR1 constant region, and UCOE EF1alpha driving PURO-BFP marker expression. Further information in *Supplementary file 7* |

*Appendix 1 Continued on next page*

*Appendix 1 Continued*

| Reagent type (species) or resource | Designation | Source or reference | Identifiers | Additional information |
|---|---|---|---|---|
| Recombinant DNA reagent | pJR104 | This paper | Addgene: 187243 | Lentiviral sgRNA vector with mU6 sgRNA promoter, CR1 constant region, and UCOE EF1alpha driving PURO-GFP marker expression. Further information in *Supplementary file 7* |
| Recombinant DNA reagent | pJR106 | This paper | Addgene: 187244 | Lentiviral sgRNA vector for CROP-seq with mU6 sgRNA promoter, CR1 constant region, and UCOE EF1alpha driving PURO-BFP marker expression. Further information in *Supplementary file 7* |
| Recombinant DNA reagent | pJR107 | This paper | Addgene: 187245 | Lentiviral sgRNA vector for CROP-seq with mU6 sgRNA promoter, CR1 constant region, and UCOE EF1alpha driving PURO-GFP marker expression. Further information in *Supplementary file 7* |
| Commercial assay or kit | Direct-zol RNA Miniprep | Zymo Research | Cat#: R2051 | |
| Commercial assay or kit | Qubit RNA Broad Range (BR) Kit | Thermo Fisher Scientific | Cat#: Q10211 | |
| Commercial assay or kit | NucleoSpin Blood kit (XL or L) | Macherey-Nagel | Cat#: 740950 (XL) or 740954 (L) | Purification of genomic DNA from cell pellets |
| Commercial assay or kit | NEBNext Ultra II Q5 PCR MasterMix | NEB | Cat#: M0544 | PCR amplification of dual-sgRNA elements from genomic DNA |
| Other | TransIT-LT1 Transfection Reagent | Mirus Bio | Cat#: MIR 2300 | Transfection reagent for lentivirus production |
| Software, algorithm | Python scripts to count dual-sgRNA elements in sequencing data and remove recombined elements | This paper | N/A | https://github.com/josephreplogle/CRISPRi-dual-sgRNA-screens, *Replogle, 2022* |
| Sequence-based reagent | Sequences of individual sgRNAs to target benchmarking genes | This paper | N/A | Sequences listed in *Supplementary file 8* |
| Sequence-based reagent | Library dJR004, Pilot genome-wide single-sgRNA human CRISPRi library | This paper | N/A | sgRNA targeting sequences in *Supplementary file 1* |
| Sequence-based reagent | Library dJR020, Pilot genome-wide dual-sgRNA human CRISRi library | This paper | N/A | sgRNA targeting sequences in *Supplementary file 1* |
| Sequence-based reagent | Library dJR072, Final genome-wide, sgRNA 1+2, dual-sgRNA human CRISPRi library with UMIs | This paper | Addgene: 187246 | sgRNA targeting sequences in *Supplementary file 4* |

*Appendix 1 Continued on next page*

*Appendix 1 Continued*

| Reagent type (species) or resource | Designation | Source or reference | Identifiers | Additional information |
|---|---|---|---|---|
| Sequence-based reagent | Library dJR073, Final genome-wide, sgRNA 3+4, dual-sgRNA human CRISPRi library with UMIs | This paper | Addgene: 187247 | sgRNA targeting sequences in *Supplementary file 4* |
| Sequence-based reagent | Library dJR074, Final genome-wide, sgRNA 5+6, dual-sgRNA human CRISPRi library with UMIs | This paper | Addgene: 187248 | sgRNA targeting sequences in *Supplementary file 4* |
| Sequence-based reagent | Library dJR075, Final genome-wide, sgRNA 1+2, dual-sgRNA human CRISPRa library with UMIs | This paper | Addgene: 187249 | sgRNA targeting sequences in *Supplementary file 9* |
| Sequence-based reagent | Library dJR076, Final genome-wide, sgRNA 3+4, dual-sgRNA human CRISPRa library with UMIs | This paper | Addgene: 187250 | sgRNA targeting sequences in *Supplementary file 9* |
| Sequence-based reagent | Library dJR077, Final genome-wide, sgRNA 5+6, dual-sgRNA human CRISPRa library with UMIs | This paper | Addgene: 187251 | sgRNA targeting sequences in *Supplementary file 9* |

