## [Editor Report]

Replogle et al. present their design of a compact and functionally validated dual sgRNA library and dCas9-effector protein that will enable new forms of CRISPRi-based screening in mammalian cells. Quantitative comparisons to previously published standards demonstrate strengths and weaknesses that along with the protocols and design strategies outlined, should enable end-users to rapidly adopt their approach.

---

## [Decision Letter]

**Decision letter after peer review:**

Thank you for submitting your article "Maximizing CRISPRi efficacy and accessibility with dual-sgRNA libraries and optimal effectors" for consideration by *eLife*. Your article has been reviewed by 2 peer reviewers, and the evaluation has been overseen by a Reviewing Editor and Didier Stainier as the Senior Editor. The following individual involved in review of your submission has agreed to reveal their identity: Mauro Calabrese (Reviewer #1).

The reviewers have discussed their reviews with one another, and the Reviewing Editor has drafted this to help you prepare a revised submission. The consensus is that it is a strong manuscript and a very useful resource for the community. The necessary revisions are mainly clarifications and should involve only minimal new experiments.

*Reviewer #1 (Public Review):*

The manuscript is written clearly and places appropriate emphasis on the strengths and weaknesses of the new approach. In Figure 1, the authors aggregate a large body of data from 126 previously performed CRISPRi screens as well as previously vetted computational predictions to create dual sgRNA libraries that can be used for succinct CRISPRi screens in human cells. They validate the functionality of their dual sgRNA library using Perturb-seq. They also note that one particular weakness of the dual sgRNA system is sgRNA recombination between lentiviruses, resulting in chimeric delivery of sgRNAs to cells. Quantitative analyses estimate that in K562 cells, recombination frequency in their new system is ~30%. In Figure 2, the authors use RNA-seq to demonstrate that in K562 cells, certain dCas9-repressors have greater off target effects than others; this was particularly striking for the SID-dCas9-Kox1 construct, which previously had been used for a handful of CRISPRi screens. In Figure 3 the authors compare the ability of different dCas9-repressors to reduce expression of several target genes, and from these experiments, identify Zim3-dCas9 as the most effective in K562 cells. In Figure 4 the authors show that Zim3-dCas9 is effective in additional human cell lines that have been used for CRISPRi screens. By my evaluation, all conclusions are well supported and justified by the data. CRISPR screens are labor and cost intensive endeavors. With this work, Replogle et al. present a validated set of novel reagents that will enable more effective and efficient CRISPR screens. The manuscript also highlights certain limitations and caveats of different forms of CRISPR screens whose clear explanation here will also benefit the community.

*Reviewer #1 (Recommendations for the authors):*

1) I appreciated the author's quantitation of recombination frequency in dual sgRNA libraries, and agree that essentially all users of the dual sgRNA approach will similarly want to discard reads that contain recombined sgRNAs. The main utility of this manuscript is the protocol it outlines and the reagents developed. In that spirit, I would request that authors provide readers with a link to a well-commented python script that enables users to repeat the author's exact protocol for detecting and discarding recombined reads.

2) Also related to protocol development and deployment: can the authors provide ranked sets of dual sgRNA libraries targeting the mouse transcriptome as well? If necessary, they could work exclusively from the predictions made in Horlbeck 2016a, but if any reasonable CRISPRi screens in mouse could serve as a validated reference, users would greatly benefit from their analysis here. CRISPR users in the mouse community would benefit, particularly for in vivo screens.

*Reviewer #2 (Public Review):*

The authors performed a series of impressive experiments to systematically establish each part of their CRISPRi method. They provided one of the most compact design of CRISPRi dual-guideRNA library, with a genome-wide coverage; they confirmed prior finding on the optimal repressor domain to generate a set of useful vectors for expressing the repressor; they showcased the usage of the system in multiple common cancer cell lines. The authors also took an important step towards providing a detailed and well-annotated protocol (in the supplementary materials) to help users of their methods. The items listed below would be helpful to further improve this work

First, while the dual guideRNA design is a useful development, the author also noted the significant rate (~30%) recombination between the two sgRNAs. This should be further discussed and evaluated in the manuscript to help readers understand the implication of this high recombination rate. For example, across replicate experiments or across cell types tested, would the recombination be stochastic, or there may be some bias of which guide would be recombined? Are there any cell-type dependencies here in terms of the recombination rate? This would also help future users to decide if they would need to check for this effect during functional screening.

Second, on the repressor development and evaluation. As the author mentioned in the text, the expression level of the repressor can confound their conclusion on fitness/efficiency comparisons of CRISPR repressor. Thus, it would be helpful to perform protein level validation using the cell lines they generated, such as a WesternBlot comparison to rule out this potential issue.

This work would also benefit from including cell proliferation/viability measurement using the selected Zim3-dCas9 repressor in multiple cell lines, as it seems this was only done initially in K562 cells. As authors noted, the fitness effects of the CRISPR repressor would be a major concern when performing functional genomics screening, so such validation of fitness-neutrality of the repressor can be very helpful for potential users of their method and approach.

Third, a major resource from this work, as the authors noted, is a suite of useful Zim3-dCas9 cell lines. The authors have performed a set of experiments to demonstrate the knockdown efficiency with dozens of guideRNAs. While this is a good initial validation, to really ensure the cell lines are performing as expected, a small scale screening in pooled fashion will be more convincing. This would be a setting more relevant for potential readers, given that pooled screening would likely be the most powerful application of these cell lines.

*Reviewer #2 (Recommendations for the authors):*

It's very helpful to have multiple constructs generated based on the Zim3-dCas9 design the author selected. It would be helpful to make a list of these constructs available in supplementary materials, and also note their design consideration or reasons to choose a particular construct, this could help readers to understand the author's perspective on how to choose the best construct for experiments.

There are some minor confusing issues, such as in Figure 3D, some of the plots don't seem to match the number/percentage given, such as the strong targeting guideRNA group where plots seem to show less repression compared with the numbers labeled, or does the number represent something different? In general, the figure labels can be improved, in particular for Figure 2A (UCOE etc. should be described in legend), Figure 3C/E, it may be better to consistently use percentage, or fraction. Figure 4C is now placed between panel A and B, it could be moved to the end.

---

## [Author Response]

Thank you for submitting your article "Maximizing CRISPRi efficacy and accessibility with dual-sgRNA libraries and optimal effectors" for consideration by eLife. Your article has been reviewed by 2 peer reviewers, and the evaluation has been overseen by a Reviewing Editor and Didier Stainier as the Senior Editor. The following individual involved in review of your submission has agreed to reveal their identity: Mauro Calabrese (Reviewer #1).The reviewers have discussed their reviews with one another, and the Reviewing Editor has drafted this to help you prepare a revised submission. The consensus is that it is a strong manuscript and a very useful resource for the community. The necessary revisions are mainly clarifications and should involve only minimal new experiments.

We thank the reviewers for their constructive feedback on our manuscript. We have revised the manuscript to address the key points that were raised and to include additional resources, as further outlined in the point-by-point response below. We are including a version of the manuscript with all changes highlighted (“track changes”) as well as a version of the manuscript in which all changes have been incorporated. We have also included a Key Resources Table.

Reviewer #1 (Public Review):The manuscript is written clearly and places appropriate emphasis on the strengths and weaknesses of the new approach. In Figure 1, the authors aggregate a large body of data from 126 previously performed CRISPRi screens as well as previously vetted computational predictions to create dual sgRNA libraries that can be used for succinct CRISPRi screens in human cells. They validate the functionality of their dual sgRNA library using Perturb-seq. They also note that one particular weakness of the dual sgRNA system is sgRNA recombination between lentiviruses, resulting in chimeric delivery of sgRNAs to cells. Quantitative analyses estimate that in K562 cells, recombination frequency in their new system is ~30%. In Figure 2, the authors use RNA-seq to demonstrate that in K562 cells, certain dCas9-repressors have greater off target effects than others; this was particularly striking for the SID-dCas9-Kox1 construct, which previously had been used for a handful of CRISPRi screens. In Figure 3 the authors compare the ability of different dCas9-repressors to reduce expression of several target genes, and from these experiments, identify Zim3-dCas9 as the most effective in K562 cells. In Figure 4 the authors show that Zim3-dCas9 is effective in additional human cell lines that have been used for CRISPRi screens. By my evaluation, all conclusions are well supported and justified by the data. CRISPR screens are labor and cost intensive endeavors. With this work, Replogle et al. present a validated set of novel reagents that will enable more effective and efficient CRISPR screens. The manuscript also highlights certain limitations and caveats of different forms of CRISPR screens whose clear explanation here will also benefit the community.Reviewer #1 (Recommendations for the authors):1) I appreciated the author's quantitation of recombination frequency in dual sgRNA libraries, and agree that essentially all users of the dual sgRNA approach will similarly want to discard reads that contain recombined sgRNAs. The main utility of this manuscript is the protocol it outlines and the reagents developed. In that spirit, I would request that authors provide readers with a link to a well-commented python script that enables users to repeat the author's exact protocol for detecting and discarding recombined reads.

We have made python scripts for the identification of recombination for dual-sgRNA libraries with or without IBCs available on github: https://github.com/josephreplogle/CRISPRi-dual-sgRNA-screens. These scripts output files counting reads aligned to all pairs and pairs excluding recombination as detailed in the documentation. They also output: (i) the number of reads that mapped to an sgRNA/IBC in the library by position and (ii) the number of reads with mapped sgRNAs that do not match the library and thus represent recombined reads. We had included a link to these scripts in the “code availability” section and have also incorporated the link in the Results section of the manuscript.

2) Also related to protocol development and deployment: can the authors provide ranked sets of dual sgRNA libraries targeting the mouse transcriptome as well? If necessary, they could work exclusively from the predictions made in Horlbeck 2016a, but if any reasonable CRISPRi screens in mouse could serve as a validated reference, users would greatly benefit from their analysis here. CRISPR users in the mouse community would benefit, particularly for in vivo screens.

We agree that dual sgRNA libraries targeting the mouse transcriptome would be helpful. As we do not have data from mouse CRISPRi/a screens to perform empirical sgRNA selection, we used the predictions from the CRISPRi/a-v2 algorithm (Horlbeck et al., *eLife* 2016, https://doi.org/10.7554/*eLife*.19760 , Supplementary file 4). We excluded sgRNAs containing BsmBI cut sites to maintain compatibility with our cloning strategy. The resulting mouse dual-sgRNA CRISPRi library is now included as Supplementary file 10 and the mouse dual-sgRNA CRISPRa library as Supplementary file 11. We also included a reference to these libraries in the Discussion section as well as a paragraph in the methods section describing the design of these libraries.

Reviewer #2 (Public Review):The authors performed a series of impressive experiments to systematically establish each part of their CRISPRi method. They provided one of the most compact design of CRISPRi dual-guideRNA library, with a genome-wide coverage; they confirmed prior finding on the optimal repressor domain to generate a set of useful vectors for expressing the repressor; they showcased the usage of the system in multiple common cancer cell lines. The authors also took an important step towards providing a detailed and well-annotated protocol (in the supplementary materials) to help users of their methods. The items listed below would be helpful to further improve this work.First, while the dual guideRNA design is a useful development, the author also noted the significant rate (~30%) recombination between the two sgRNAs. This should be further discussed and evaluated in the manuscript to help readers understand the implication of this high recombination rate. For example, across replicate experiments or across cell types tested, would the recombination be stochastic, or there may be some bias of which guide would be recombined? Are there any cell-type dependencies here in terms of the recombination rate? This would also help future users to decide if they would need to check for this effect during functional screening.

We agree that recombination is an important limitation of dual-sgRNA screens. We included additional analyses and data in the revised manuscript to help readers understand the implications of the observed recombination.

First, we performed growth screens using dual-sgRNA libraries in two additional cell lines (RPE1 and Jurkat) to address the potential cell type specificity of lentiviral recombination. We cloned a dual-sgRNA library targeting DepMap Common Essential genes (n=2291 dual-sgRNA elements). We transduced cells with this library, harvested cells at day 7 post-transduction, amplified sgRNA cassettes from extracted genomic DNA, and sequenced to quantify sgRNA recombination rates. We found similar recombination rates of dual-sgRNA constructs isolated from these three cell types (observed K562 recombination rate 29%; observed RPE1 recombination rate 26%; observed Jurkat recombination rate 24%).

Next, we compared the recombination rates of each dual-sgRNA element. Our expectation was that lentiviral recombination would be largely stochastic per element based on the known mechanism of lentiviral recombination previously discussed in Adamson et al. 2018 (https://www.biorxiv.org/content/10.1101/298349v1.full) given that the constant region between sgRNAs (400bp) far exceeds the length of sgRNA targeting regions (20bp). However, we would also expect apparent recombination rates to be artificially inflated for dual-sgRNAs with strong growth phenotypes, as the stronger growth phenotypes of unrecombined dual-sgRNAs compared to recombined dual-sgRNAs will lead to dropout of unrecombined dual-sgRNAs. To account for this bias, we began by comparing the recombination rate for non-targeting control dual-sgRNAs excluding those with growth phenotypes across replicates of our K562 screens. There was only a weak correlation between the recombination rate for non-targeting control dual-sgRNAs (*r* = 0.30; Figure 1 – Figure Supplement 1E). We next compared the recombination rates of all dual-sgRNA elements (both targeting and non-targeting) across replicates of our K562 screens. As expected, we observed that the recombination rate of elements was correlated across replicates (*r* = 0.77; Figure 1 – Figure Supplement 1F), and the recombination rate was strongly anticorrelated with the growth phenotype of dual-sgRNAs in K562 cells (*r* = -0.84; Figure 1 – Figure Supplement 1G). We have integrated these data into the manuscript.

Second, on the repressor development and evaluation. As the author mentioned in the text, the expression level of the repressor can confound their conclusion on fitness/efficiency comparisons of CRISPR repressor. Thus, it would be helpful to perform protein level validation using the cell lines they generated, such as a WesternBlot comparison to rule out this potential issue.

We agree that differences in expression levels of the effectors can confound comparisons and that Western Blotting for such differences would be valuable. That said, any such analyses would not substantively alter the main claim of our paper, which is that Zim3-dCas9 provides excellent on-target knockdown in the absence of non-specific effects on cell growth or gene expression. This finding is of immediate practical use to the community. By no means are we claiming that we eliminated all possible confounding factors nor do we think that it is possible to do so. To avoid overstating our findings, we had acknowledged in the discussion that expression levels may indeed be a confounding factor, we had noted in the methods section that the dCas9-MeCP2 effector uses a different coding sequence for dCas9, which may contribute to differences in expression, and we had noted that other effectors may prove useful in some settings. We have further emphasized that differences in expression levels may contribute to our results in the revised manuscript.

This work would also benefit from including cell proliferation/viability measurement using the selected Zim3-dCas9 repressor in multiple cell lines, as it seems this was only done initially in K562 cells. As authors noted, the fitness effects of the CRISPR repressor would be a major concern when performing functional genomics screening, so such validation of fitness-neutrality of the repressor can be very helpful for potential users of their method and approach.

To address this point, we assessed the proliferation of HepG2, HuTu-80, and HT29 cells expressing Zim3-dCas9. Expression of Zim3-dCas9 did not have a negative impact on proliferation in any of these cell types, providing further evidence that the Zim3-dCas9 will be broadly useful. We included these data in Figure 4 – Figure Supplement 2 in the revised manuscript. That said, we cannot rule out that expression of Zim3-dCas9 may be detrimental in other cell types. Indeed, we want to emphasize that users should evaluate both on-target knockdown and lack of non-specific effects of effectors in new cell models before proceeding to large-scale experiments. The assays and protocols we describe are ideally suited for this purpose. We have further emphasized this point in the discussion section to guide users.

Third, a major resource from this work, as the authors noted, is a suite of useful Zim3-dCas9 cell lines. The authors have performed a set of experiments to demonstrate the knockdown efficiency with dozens of guideRNAs. While this is a good initial validation, to really ensure the cell lines are performing as expected, a small scale screening in pooled fashion will be more convincing. This would be a setting more relevant for potential readers, given that pooled screening would likely be the most powerful application of these cell lines.

While conducting the work described in this manuscript, we had used the Zim3-dCas9 RPE1 cell line for a large-scale pooled screen with single-cell RNA-seq readout (Perturb-seq, Replogle et al. 2022). Across greater than 2000 target genes, the median knockdown was 91.6%, which provides strong validation that Zim3-dCas9 performs as expected in this cell line. We had noted this point in the discussion section of our manuscript.

Reviewer #2 (Recommendations for the authors):It's very helpful to have multiple constructs generated based on the Zim3-dCas9 design the author selected. It would be helpful to make a list of these constructs available in supplementary materials, and also note their design consideration or reasons to choose a particular construct, this could help readers to understand the author's perspective on how to choose the best construct for experiments.

We had included a list of constructs in Supplementary file 7. We had also included considerations for selecting constructs in our cell line generation protocol. To make it easier for readers to find these protocols, we have made them publicly available at a separate location (https://www.jostlab.org/resources/ and https://weissman.wi.mit.edu/resources/) and we have included more explicit references to the protocols in the result section.

There are some minor confusing issues, such as in Figure 3D, some of the plots don't seem to match the number/percentage given, such as the strong targeting guideRNA group where plots seem to show less repression compared with the numbers labeled, or does the number represent something different? In general, the figure labels can be improved, in particular for Figure 2A (UCOE etc. should be described in legend), Figure 3C/E, it may be better to consistently use percentage, or fraction. Figure 4C is now placed between panel A and B, it could be moved to the end.

We thank the reviewer for bringing up these issues. For Figure 3D, the numbers do match the plots, because we quantified knockdown at the protein level using the median antibody staining signal. Although using median signal overestimates knockdown when knockdown is bimodal, as is the case for some of our samples, we believe that median values are the appropriate metric for flow cytometric quantification of protein levels. We now more clearly emphasize the quantification metric in the figure legend.

In addition, we updated the legend for Figure 2A to include a description of the abbreviations, we updated Figure 3 to consistently depict percentages, and we re-arranged Figure 4.